# Efficient coding of natural scene statistics predicts discrimination thresholds for grayscale textures

**Tiberiu Tesileanu[1]\*, Mary M Conte[2], John J Briguglio[3], Ann M Hermundstad[3], Jonathan D Victor[2], Vijay Balasubramanian[4]**

[1]Flatiron Institute, New York, United States; [2]Feil Family Brain and Mind Institute, Weill Cornell Medical College, New York, United States; [3]Janelia Research Campus, Ashburn, United States; [4]David Rittenhouse Laboratories, University of Pennsylvania, Philadelphia, United States

**Abstract** Previously, in Hermundstad et al., 2014, we showed that when sampling is limiting, the efficient coding principle leads to a 'variance is salience' hypothesis, and that this hypothesis accounts for visual sensitivity to binary image statistics. Here, using extensive new psychophysical data and image analysis, we show that this hypothesis accounts for visual sensitivity to a large set of grayscale image statistics at a striking level of detail, and also identify the limits of the prediction. We define a 66-dimensional space of local grayscale light-intensity correlations, and measure the relevance of each direction to natural scenes. The 'variance is salience' hypothesis predicts that two-point correlations are most salient, and predicts their relative salience. We tested these predictions in a texture-segregation task using un-natural, synthetic textures. As predicted, correlations beyond second order are not salient, and predicted thresholds for over 300 second-order correlations match psychophysical thresholds closely (median fractional error <0.13).

**\*For correspondence:**
ttesileanu@gmail.com

**Competing interests:** The authors declare that no competing interests exist.

## Introduction

Neural circuits in the periphery of the visual (*Laughlin, 1981*; *Atick and Redlich, 1990*; *van Hateren, 1992*; *Fairhall et al., 2001*; *Ratliff et al., 2010*; *Liu et al., 2009*; *Borghuis et al., 2008*; *Garrigan et al., 2010*; *Kuang et al., 2012*), auditory (*Schwartz and Simoncelli, 2001*; *Lewicki, 2002*; *Smith and Lewicki, 2006*; *Carlson et al., 2012*), and perhaps also olfactory (*Teşileanu et al., 2019*) systems use limited resources efficiently to represent sensory information by adapting to the statistical structure of the environment (*Sterling and Laughlin, 2015*). There is some evidence that this sort of efficient coding might also occur more centrally, in the primary visual cortex (*Olshausen and Field, 1996*; *Bell and Sejnowski, 1997*; *Vinje and Gallant, 2000*; *van Hateren and van der Schaaf, 1998*) and perhaps also in the entorhinal cortex (*Wei et al., 2015*). Behaviorally, efficient coding implies that the threshold for perceiving a complex sensory cue, which depends on the collective behavior of many cells in a cortical circuit, should be set by its variance in the natural environment. However, the nature of this relationship depends on the regime in which the sensory system operates. Specifically, in conventional applications of efficient coding theory where sampling is abundant, high variance is predicted to be matched by high detection thresholds. The authors of *Hermundstad et al., 2014* argued instead that texture perception occurs in a regime where sampling noise is the limiting factor. This leads to the *opposite* prediction (*Tkacik et al., 2010*; *Doi and Lewicki, 2011*; *Hermundstad et al., 2014*), namely that high variance should lead to a *low* detection threshold, summarized as *variance is salience* (*Hermundstad et al., 2014*). Tests of this prediction in *Tkacik et al., 2010*; *Hermundstad et al., 2014* showed that it holds for the visual detection of simple black-and-white binary textures.

These binary textures, while informative about visual sensitivity, are a highly restricted set and do not capture many perceptually-salient properties of natural scenes. Moving to a complete description of visual textures, however, requires specifying the co-occurrence of all possible patterns of light across a visual image, and is generally intractable. One way to make this specification tractable is to construct a local and discretized grayscale texture space, in which luminance is drawn from a discrete set and correlations in luminance are only specified up to a given neighborhood size. For example, if we consider four spatially-contiguous squares ('checks') with binary intensities, there are $2^4 = 16$ patterns that can each occur with different probabilities in a given texture. Imposing translation symmetry constrains these 16 parameters, leading to a 10-dimensional space of textures (*Hermundstad et al., 2014*). This space can be explored by synthesizing artificial binary textures with prescribed combinations of parameters (*Victor et al., 2005*; *Victor and Conte, 2012*), and by analyzing the relative contributions of these parameters to the correlated structure of natural images (*Tkacik et al., 2010*; *Hermundstad et al., 2014*). Here, we generalized these synthesis and analysis methods to multiple gray levels and used this to probe efficient encoding of grayscale textures composed of correlated patterns of three luminance levels (dark, gray, light) specified within blocks of four contiguous checks. We chose to add only one intermediate gray level compared to the binary case because it is the simplest generalization of binary textures that allows us to explore perceptual sensitivity to grayscale textures. Because the number of possible visual patterns increases as a power law in the number of distinguishable luminance values, this generalization already yields a very high-dimensional space: for intensities constrained to $G = 3$ discrete values, there are $G^4 = 81$ patterns of four checks with three gray levels, leading to a 66-dimensional space of textures after accounting for the constraints of translation invariance.

This grayscale texture space enabled us to probe and interpret the relationship between natural scene statistics and psychophysics in much greater detail than is possible with binary textures. In particular, the 'variance is salience' hypothesis qualitatively predicts that directions corresponding to two-point correlations will be most perceptually salient, and it quantitatively predicts detection thresholds in different directions of this salient part of the texture space. (Two coordinates corresponding to contrast, which are also highly salient, are zeroed out by the preprocessing in our natural image analysis; we therefore do not probe these directions.) We tested these predictions by asking observers to report the location of textured strips presented rapidly against a background of white noise, and we found detailed agreement with the theory. By further exploiting symmetries in the distribution of grayscale textures, we show that human behavior not only reflects the relative informativeness of natural visual textures, but it also parallels known invariances in natural scenes. Natural scenes also have a notable, previously studied, asymmetry between bright and dark (*Ratliff et al., 2010*; *Tkacik et al., 2010*) which is reflected in the anatomy and physiology of visual circuits, and in visual behavior (*Ratliff et al., 2010*; *Tkacik et al., 2010*; *Zemon et al., 1988*; *Chubb et al., 2004*; *Jin et al., 2008*; *Komban et al., 2014*; *Kremkow et al., 2014*). The asymmetry is rooted ultimately in the lognormal distribution and spatial correlation structure of light intensities in natural scenes (*Ratliff et al., 2010*; *Tkačik et al., 2011*). Our image processing pipeline (see Materials and methods) starts by taking the logarithm of the pixel intensities and removing the large-scale $1/f$ spatial correlations, and thus significantly reduces the dark-light asymmetry. The ternarization procedure that we then employ further diminishes the asymmetry, allowing us to focus on other aspects of natural scene statistics.

## Results

### Local textures with multiple gray levels

We define textures in terms of statistical correlations between luminance levels at nearby locations, generalizing the methods developed for binary images (*Victor and Conte, 2012*; *Hermundstad et al., 2014*) to three luminance levels. If we consider four 'checks' arranged in a $2 \times 2$ square, the three luminance levels lead to $3^4 = 81$ possible patterns, and their frequency of occurrence in an image is equivalently parameterized by intensity correlations within the square. Thus there is an 81-dimensional space of ternary textures defined by correlations within square

arrangements of checks. However, translation invariance constrains these 81 probabilities, reducing the number of independent statistics and thus the dimension of the texture space.

We can quantify the statistics of such textures in an image patch by gliding a $2 \times 2$ block (a 'glider') over the patch and analyzing the luminance levels at different locations within this block (*Figure 1A*). At the most basic level, we can measure the luminance histogram at each of the four check locations in the glider. Check intensities can take three values (0, 1, or 2, for black, gray, or white), and the corresponding frequencies of occurrence must add to one, leaving two free parameters. If the histograms at each of the four locations within the glider were independent, this would lead to $4 \times 2 = 8$ texture dimensions. However, because of translation invariance in natural images, the luminance histograms at each location must be the same, leaving only two independent dimensions of texture space from the single-check statistics.

Next, we can analyze the statistics of luminance levels at pairs of locations within the glider. Taking into account translation invariance, there are four ways to position these pairs (*Figure 1B*), each with a different orientation. For each orientation, we can calculate either the sum $A + B$ or the difference $A - B$ of the luminance values $A$ and $B$ at the two locations. This yields eight possible texture *groups* (*Figure 1B*). Within each group, we build texture coordinates by counting the fraction of occurrences in which $A \pm B$ is equal to 0, 1, or 2, up to a multiple of 3; that is, we are building a histogram of $A \pm B$ *modulo* 3, here denoted $\mathrm{mod}(A \pm B, 3)$. The appearance of the modulo function here

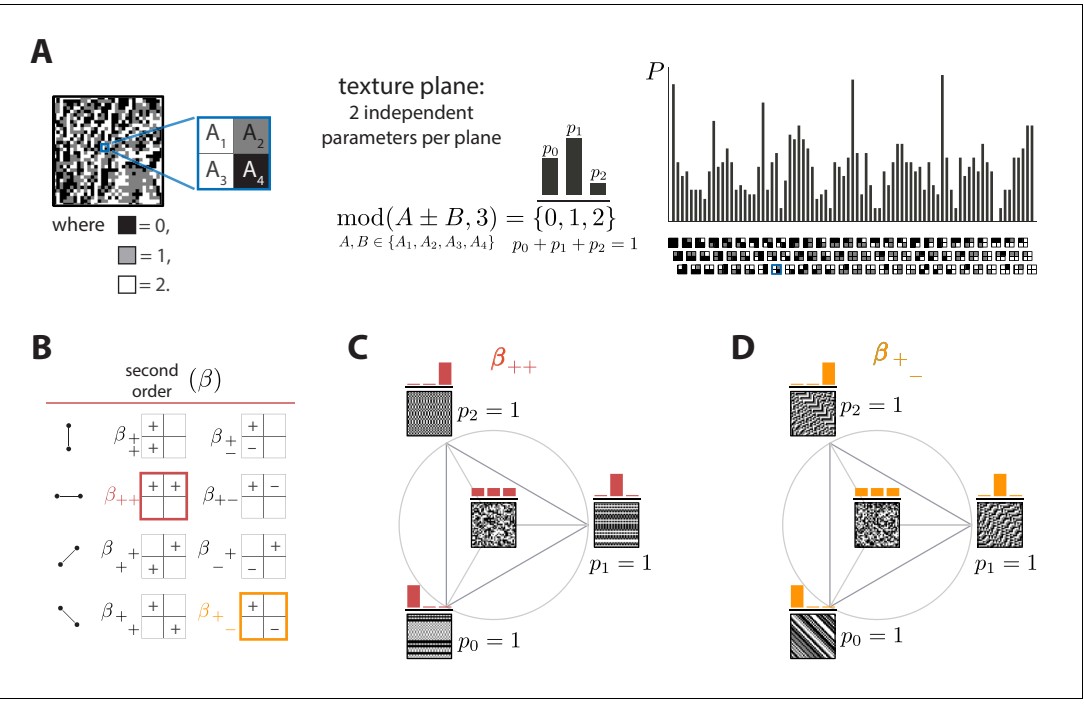

**Figure 1.** Ternary texture analysis. (A) With three luminance levels there are $3^4 = 81$ possible check configurations for a $2 \times 2$ block (histogram on the right). We parametrize the pairwise correlations within these blocks using modular sums or differences of luminance values at nearby locations, $\mathrm{mod}(A \pm B, 3)$ (see Materials and methods and Appendix 1 for details). This notation denotes the remainder after division by 3, so that for example $\mathrm{mod}(4, 3) = 1$ and $\mathrm{mod}(-1, 3) = 2$. The texture coordinates are defined by the probabilities $p_0$, $p_1$, $p_2$ with which $\mathrm{mod}(A \pm B, 3)$ equals its three possible values, 0, 1, or 2. These three probabilities must sum to 1, so there are only two independent coordinates for each triplet of probabilities. (B) The eight second-order groups (planes) of texture coordinates in the ternary case. A texture group is identified by the choice of orientation of the pair of checks for which the correlation is calculated, and by whether a sum or a difference of luminance values is used. The greek letter notation ($\beta$ for the second-order planes) mirrors the notation used in *Hermundstad et al., 2014*. (C and D) Example texture groups ('simple' planes). The origin is the point $p_0 = p_1 = p_2 = 1/3$, representing an unbiased random texture. The interior of the triangle shows the allowed range in the plane where all the probability values are non-negative. The vertices are the points where only one of the probabilities is nonzero. An example texture patch is shown for the origin, as well as for each of the vertices of the probability space.

is a consequence of using a Fourier transform in the space of luminance levels, which is convenient for incorporating translation invariance into our coordinate system (see Materials and methods and Appendix 1). The fractions of values of $\mathrm{mod}(A+B, 3)$ equal to 0, 1, or 2 must add up to 1, so that each texture group is characterized by two independent coordinates, that is, a plane in texture space. These 8 planes constitute 16 independent dimensions of the texture space, in addition to the two dimensions needed to capture the histogram statistics at individual locations.

We can similarly analyze the joint statistics of luminance levels at three checks within the glider, or all four together. There are four ways to position 3-check gliders within the $2 \times 2$ square, and, for each of these positions, eight parameters are needed to describe their occurrence frequencies, once the first- and second-order statistics have been fixed. This leads to $4 \times 8 = 32$ third-order parameters. For configurations of all four checks, 16 parameters are required, once first-, second-, and third-order parameters are fixed. These $32 + 16 = 48$ parameters, in addition to the 18 parameters described above, lead to a 66-dimensional texture space. This provides a complete parameterization of the $2 \times 2$ configurations with three gray levels. See Methods and Appendix 1 for a detailed derivation and a generalization to higher numbers of gray levels.

In order to probe human psychophysical sensitivity to visual textures, we need an algorithm for generating texture patches at different locations in texture space. To do so, we use an approach that generalizes the methods from *Victor and Conte, 2012*. Briefly, we randomly populate entries of the first row and/or column of the texture patch, and we then sequentially fill the rest of the entries in the patch using a Markov process that is designed to sample from a maximum-entropy distribution in which some of the texture coordinates are fixed (see Methods and Appendix 2 for details). Examples of texture patches obtained by co-varying coordinates within a single texture group are shown in *Figure 1C and D*. Examples of textures obtained by co-varying coordinates in two texture groups are shown in *Figure 2*. We refer to the first case as 'simple' planes, and the second as 'mixed' planes.

When applying these methods to the analysis of natural images, we bin luminance values to produce equal numbers of black, white, and gray checks (details below), thus equalizing the previously studied brightness statistics in scenes (e.g., *Zemon et al., 1988*; *Chubb et al., 2004*; *Jin et al., 2008*; *Ratliff et al., 2010*; *Tkacik et al., 2010*; *Tkačik et al., 2011*; *Komban et al., 2014*; *Kremkow et al., 2014*). This procedure allowed us to focus on higher-order correlations. In previous work, we found that median-binarized natural images show the highest variability in pairwise correlations, and that observers are correspondingly most sensitive to variations in these statistics (*Tkačik et al., 2011*; *Hermundstad et al., 2014*). In view of this, we focused our attention on pairwise statistics. For three gray levels, these comprise a 16-dimensional 'salient' subspace of the overall texture space.

## Natural image statistics predict perceptual thresholds
### Predictions from natural scene statistics
The 'variance is salience' hypothesis from *Hermundstad et al., 2014* predicts that the most salient directions in texture space (for which detection thresholds are low) will be those along which there is the highest variance across image patches, while the least salient directions (for which detection thresholds are high) will be those with lowest variance. To test these predictions, we first mapped natural image patches to their corresponding locations within the texture space (as in *Hermundstad et al., 2014*). We then computed the inverse of the standard deviation of the natural image distribution along each direction, and we used this as our prediction of detection thresholds. The procedure is sketched in *Figure 3* and described in more detail in Methods. We found that natural images have much higher variance in the second-order coordinate planes than in the third- and fourth-order planes. This predicted that textures exhibiting variability in the second-order planes would be most salient to observers, and thus would be amenable to a quantitative comparison between theory and behavior. Thus, we computed the predicted detection thresholds in four single and twenty-two mixed second-order coordinate planes, and we scaled this set of thresholds by a single overall scaling factor that was chosen to best match the behavioral measurements (blue dots in *Figure 4C* and *Figure 5*).

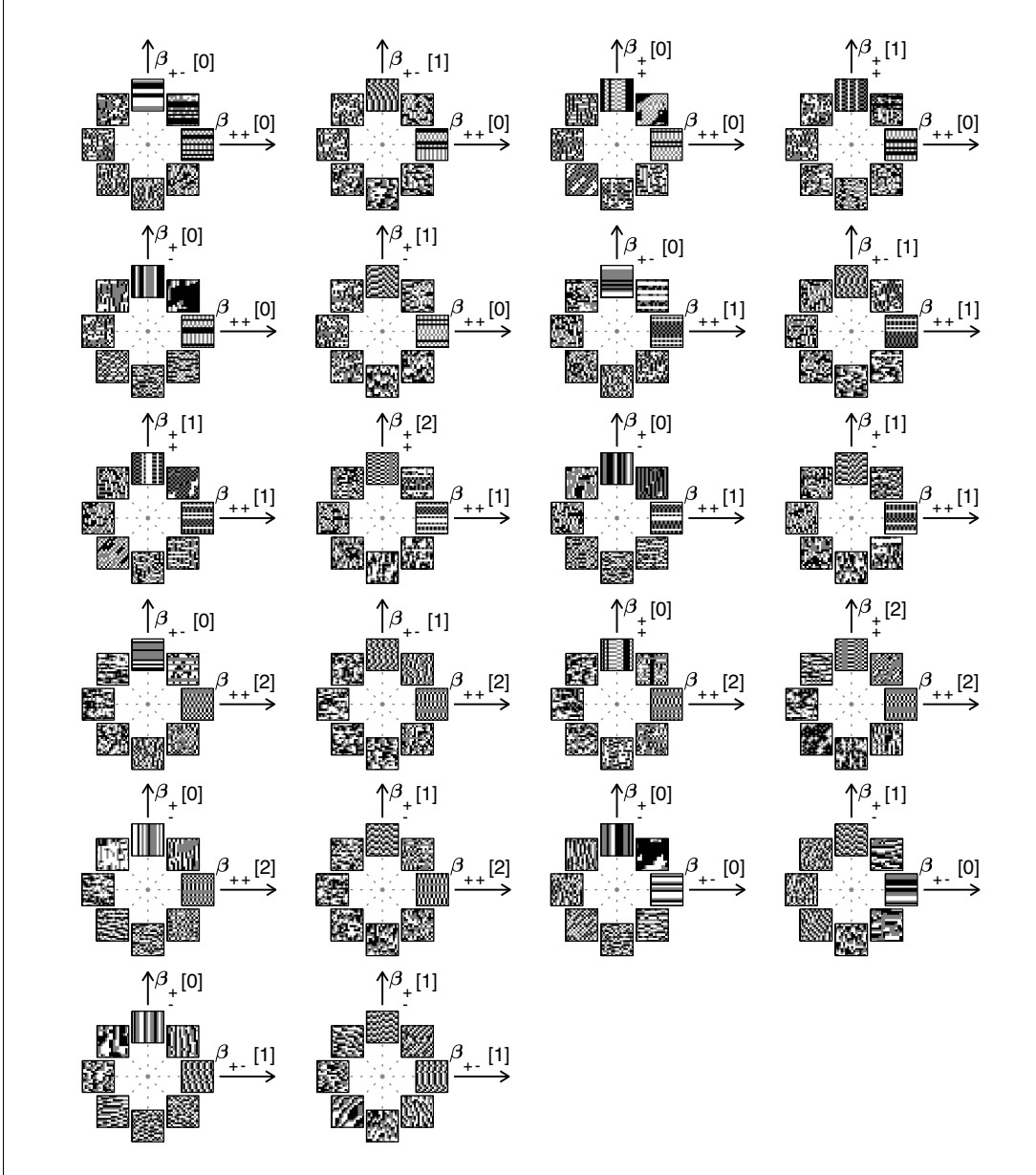

**Figure 2.** Examples of textures from all the mixed planes for which psychophysics data are available. Each patch is obtained by choosing coordinates in two different texture groups: for instance, a point in the $(\beta_{++}[0], \beta_{+-}[1])$ plane (row 1, column 2) corresponds to choosing the probabilities that $\mathrm{mod}(A + B, 3) = 0$ and $\mathrm{mod}(A - B, 3) = 1$. Apart from these constraints, the texture is generated to maximize entropy (see Materials and methods and Appendix 2). The center of the coordinate system in all planes corresponds to an unbiased texture (i.e., the probability for each direction is 1/3), while a mixed-plane coordinate equal to one corresponds to full saturation (i.e., the probability for that direction is 1). The dashed lines indicate the directions in texture space along which the illustrated patches were generated. The patches within a given plane are drawn at a constant distance from the center, but the precise amount of texture saturation varies, according to the largest saturation that could be generated in each direction. Note that along some directions, the maximum saturation is limited by the way in which the texture coordinates are defined, or by the texture synthesis procedure (see Appendix 2).

## Psychophysical measurements

To measure the sensitivity of human subjects to different kinds of textures, we used a four-alternative forced-choice paradigm following *Hermundstad et al., 2014*; *Victor and Conte, 2012*; *Victor et al., 2013*; *Victor et al., 2015*. Subjects were briefly shown an array in which a rectangular

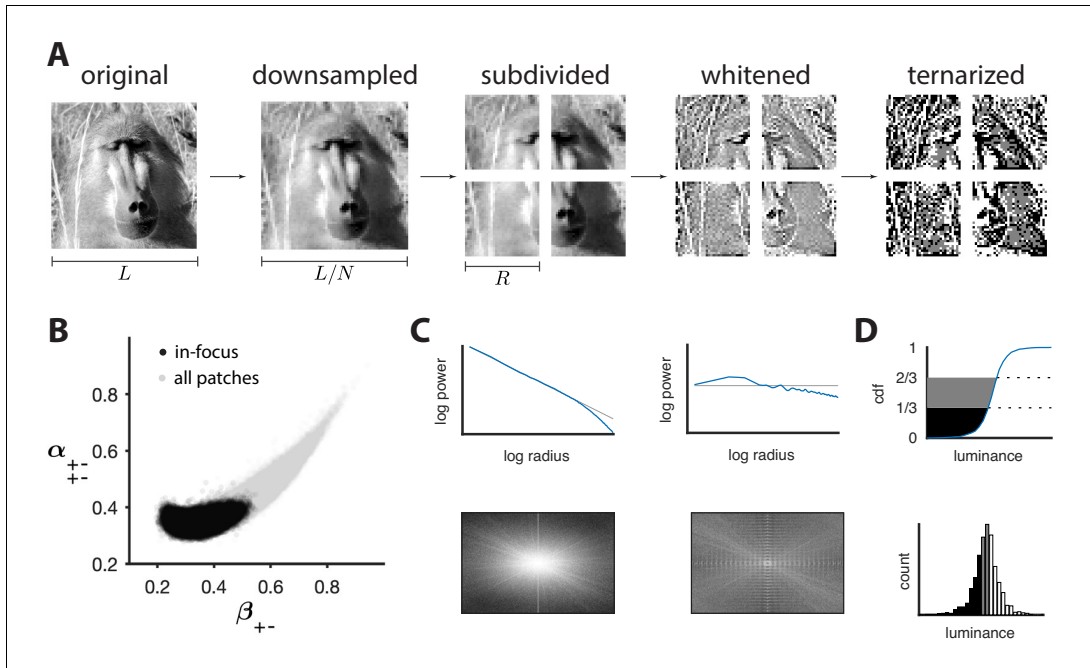

**Figure 3.** Preprocessing of natural images. (A) Images (which use a logarithmic encoding for luminance) are first downsampled by a factor $N$ and split into square patches of size $R$. The ensemble of patches is whitened by applying a filter that removes the average pairwise correlations (see panel C), and finally ternarized after histogram equalization (see panel D). (B) Blurry images are identified by fitting a two-component Gaussian mixture to the full distribution of image textures (shown in light gray). This is shown here in a particular projection involving a second-order direction ($\beta_{+-}$) and a fourth-order one ($\alpha_{+-}^{+-}$). The texture analysis is restricted to the component with higher contrast, which is shown in black on the plot. Note that a value of 1/3 on each axis corresponds to the origin of the texture space. (C) Power spectrum before and after filtering an image from the dataset. (D) Images are ternarized such that within each patch a third of the checks are converted to black, a third to gray, and a third to white. The processing pipeline illustrated here extends the analysis of *Hermundstad et al., 2014* to multiple gray levels.

strip positioned near the left, right, top, or bottom edge was defined by a texture difference: either the strip was structured and the background was unstructured, or the background was structured and the strip was unstructured. Structured patterns were constructed using the texture generation method described above, and unstructured patterns were generated by randomly and independently drawing black, gray, or white checks with equal probability. The texture analysis procedure in *Figure 3* included a whitening step that removed long-range correlations in natural images, and the local textures generated to test the predictions psychophysically also lacked these correlations. This is appropriate, because, during natural vision, dynamic stimuli engage fixational eye movements that whiten the visual input (*Rucci and Victor, 2015*). Spatial filtering has also been thought to play a role in whitening (*Atick and Redlich, 1990*), but in vitro experiments (*Simmons et al., 2013* found that adaptive spatiotemporal receptive field processing did not by itself whiten the retinal output, but rather served to maintain a similar degree of correlation across stimulus conditions). We infer from these studies that short visual stimuli like our 120 ms presentations should be pre-whitened, to make up for the absence of fixational eye movements that produce whitening in natural, continuous viewing conditions.

Subjects were asked to indicate the position of the differently textured strip within the array (*Figure 4A*). Thresholds were obtained by finding the value of a texture coordinate for which the subjects' performance was halfway between chance and perfect (*Figure 4B*; see Materials and methods for details). For the second-order planes, subjects were highly consistent in their relative sensitivity to different directions in texture space, with a single scaling factor accounting for a majority of the inter-subject variability (see Appendix 6). The subject-average thresholds in the second order planes are shown in *Figure 4C* and *Figure 5* (red crosses and error bars). As predicted by the natural

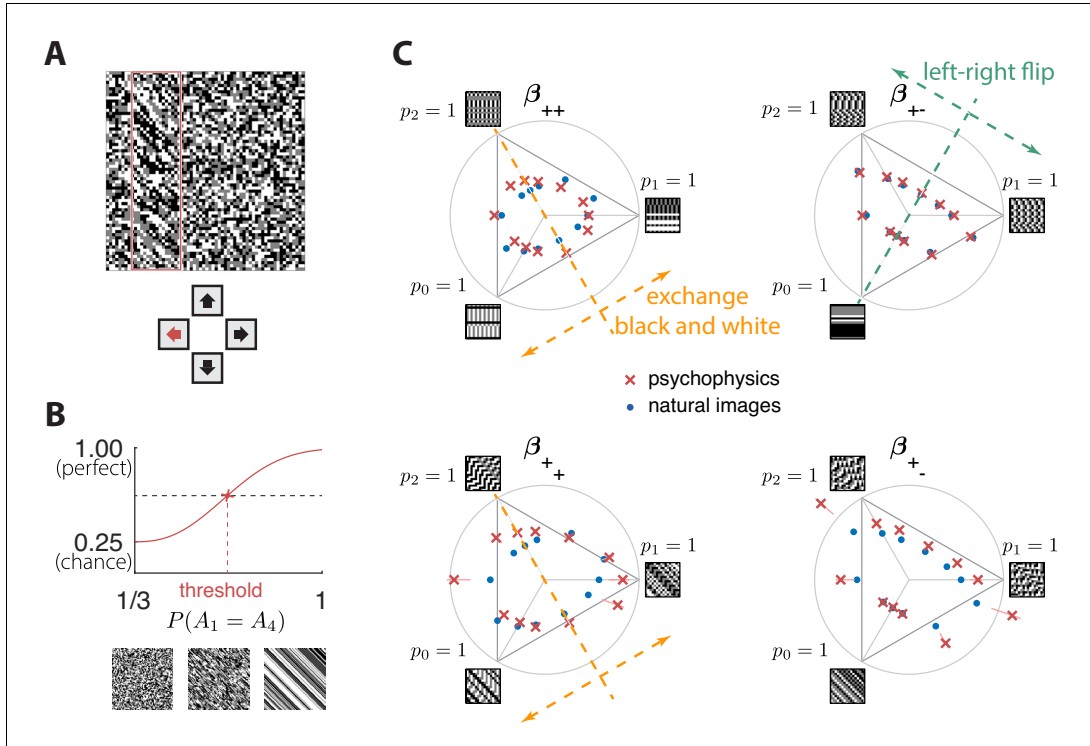

**Figure 4.** Experimental setup and results in second-order simple planes. (**A**) Psychophysical trials used a four-alternative forced-choice task in which the subjects identified the location of a strip sampled from a different texture on top of a background texture. (**B**) The subject's performance in terms of fraction of correct answers was fit with a Weibull function and the threshold was identified at the mid-point between chance and perfect performance. Note that if the subject's performance never reaches the mid-point on any of the trials, this procedure may extrapolate a threshold that falls outside the valid range for the coordinate system (see, e.g., the points outside the triangles in panel **C**). This signifies a low-sensitivity direction of texture space. (**C**) Measured thresholds (red crosses with pink error bars; the error bars are in most cases smaller than the symbol sizes) and predicted thresholds (blue dots) in second-order simple planes. Thresholds were predicted to be inversely proportional to the standard deviation observed in each texture direction in natural images. The plotted results used downsampling factor $N = 2$ and patch size $R = 32$. A single scaling factor for all planes was used to match to the psychophysics. The orange and green dashed lines show the effect of two symmetry transformations on the texture statistics (see text).

scene analysis, sensitivity in the third and fourth-order planes was low; in fact, detection thresholds could not be reliably measured in most directions beyond second order (Appendix 6).

## Variance predicts salience

Predicted detection thresholds were in excellent agreement with those measured experimentally (*Figure 4C* and *Figure 5*), with a median absolute log error of about 0.13. Put differently, 50% of the measurements have relative errors below 13%, since in this regime, the log error is very well approximated by relative error (see Materials and methods and Appendix 4). This match is unlikely to be due to chance—a permutation test yields $p < 10^{-4}$ for the hypothesis that all measured thresholds were drawn independently from a single distribution that did not depend on the texture direction in which they were measured (see Materials and methods and Appendix 4 for details and further statistical tests; 95% of the 10,000 permutation samples exhibited median absolute log errors in the range [0.24, 0.31]). The comparison between theory and experiment was tested in 12 directions in each of 26 single and mixed planes, for a total of 311 different thresholds (the measurement uncertainty in one direction in the $\beta_{++}[0]; \beta_{+}^{+}[1]$ mixed plane was too large, and we discarded that datapoint); a single scaling factor was used to align these two sets of measurements. Note that these measurements are not fully independent: the natural image predictions within each plane lie on an

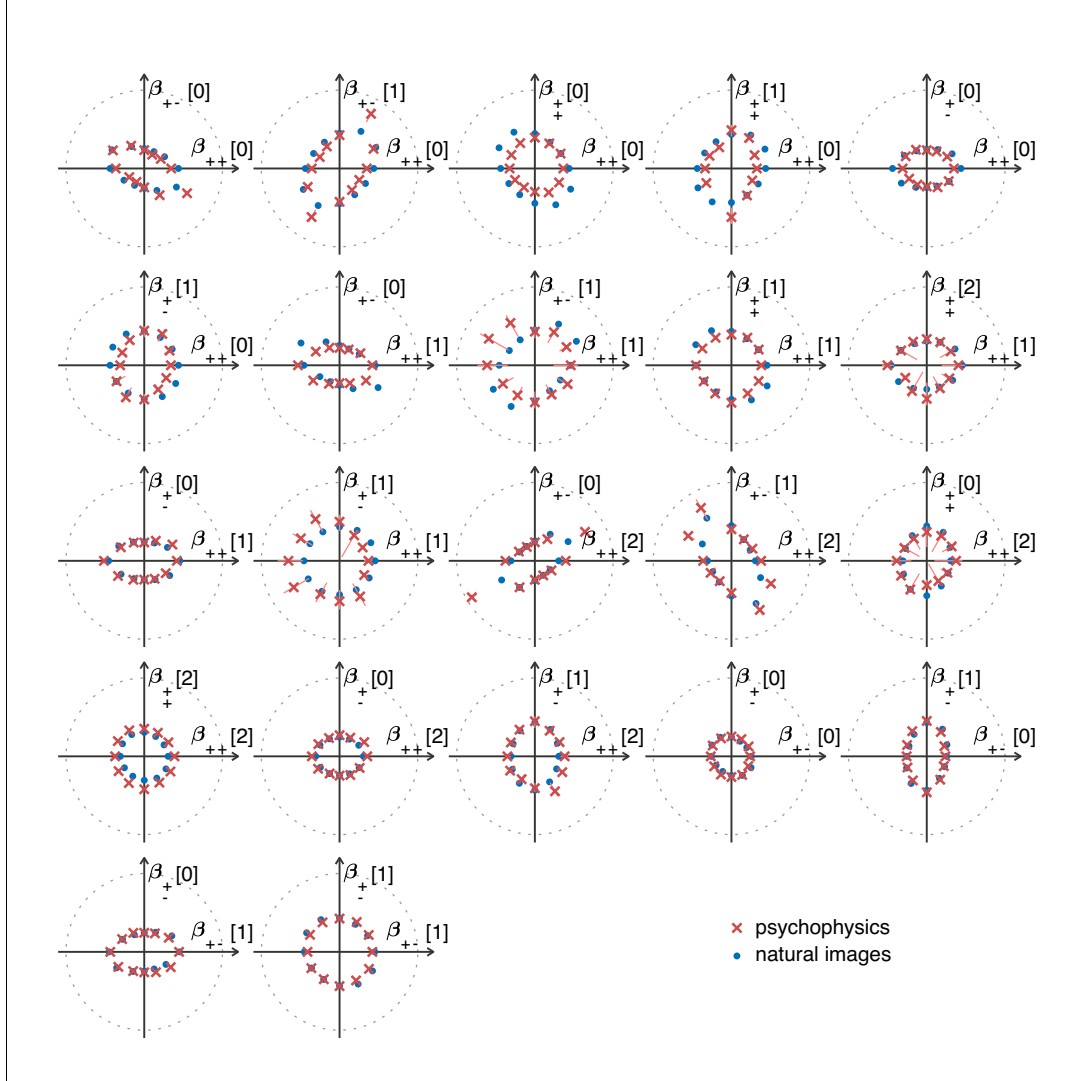

**Figure 5.** The match between measured (red crosses and error bars) and predicted (blue dots) thresholds in 22 mixed planes. Each plot corresponds to conditions in which the coordinates in two different texture groups are specified, according to the axis labels. For instance, column two in row one is the $(\beta_{++}[0], \beta_{+-}[1])$ plane; the two coordinates correspond to choosing the probabilities that $\mathrm{mod}(A + B, 3) = 0$ and $\mathrm{mod}(A - B, 3) = 1$. As in *Figure 2*, the center of the coordinate system in these planes corresponds to an unbiased texture (i.e., the probability for each direction is 1/3), while a coordinate equal to 1—indicated by the gray dotted circle— corresponds to full saturation (i.e., the probability for that direction is 1).

ellipse by construction; the psychophysical thresholds are measured independently at each point but are generally well-approximated by ellipses. Even taking this into account, the match between the predictions and the data is unlikely to be due to chance ($p<10^{-4}$; 95% range for the median absolute log error [0.16, 0.22]; see Appendix 4).

However, not all thresholds are accurately predicted from natural images. While some of the mismatches seem random and occur for directions in texture space where the experimental data have large variability (pink error bars in *Figure 4C* and *Figure 5*), there are also systematic discrepancies. Natural-image predictions tend to underestimate the threshold in the simple planes (median log prediction error −0.090) and overestimate the threshold in mixed planes (median log prediction error +0.008). That is, in human observers, detection of simultaneously-present multiple correlations is disproportionately better than predicted from the detection thresholds for single correlations, to a mild degree (see Appendix 5 for details).

A second observation is that prediction errors tend to be larger for the sum-correlations (such as $\beta_{++}$) than for the difference-correlations (such as $\beta_{+-}$), independent of the direction of the error. This may be a consequence of the way that modular arithmetic and gray-level discretization interact, leading to a kind of non-robustness of the sum-correlations. Specifically, the most prominent feature of the patterns induced by the sum-correlations is that there is a single gray level that occurs in runs (e.g., $p_0 = 1$ leads to runs of black checks, $p_1 = 1$ leads to runs of white checks, and $p_2 = 1$ leads to runs of gray checks; see *Figure 1C*). On the other hand, for difference correlations, $p_0 = 1$ leads to runs of all gray levels, while $p_1 = 1$ and $p_2 = 1$ lead to 'mini-gradients' that cycle between white, gray, and black (*Figure 1D*). Thus, the sum-correlations are subject to the particular assignment of gray levels and modulus while the difference-correlations rely on relationships that hold independent of these choices.

Independent of these trends, the natural-image predictions tend to underestimate the thresholds in directions with very low variance even while they match the thresholds in directions with high variance (see Appendix 5). This suggests the need to go beyond the linear efficient-coding model employed here. A simple generalization that interpolates between existing analytically solvable models (*Hermundstad et al., 2014*) involves a power-law transformation of the natural-image variances, $\mathrm{threshold} \propto (\mathrm{standard\ deviation})^{-\eta}$. We use a default exponent of $\eta = 1$ throughout the text. The exponent $\eta$ that best matches our data is close but probably not equal to 1 (the 95% credible interval is [0.81, 0.98]; see Appendix 4), suggesting that a weak power-law nonlinearity might be involved. The inferred range for $\eta$ also confirms that the measured thresholds are not independent of the predicted ones (which would have mapped to $\eta = 0$).

In sum, the modest systematic discrepancies between the natural-image predictions and the measured thresholds indicate that the efficient-coding model has limitations, and the observed mismatches can guide future studies that go beyond this model. More generally, we do not expect the predictions of efficient coding to hold indefinitely: adapting to increasingly precise environmental statistics must ultimately become infeasible, both because of insufficient sampling, and because of the growing computational cost required for adaptation. Whether, and to what extent, these issues are at play is a subject for future work.

These results are robust to several variations in our analysis procedure. We obtain similar results when we either vary the sub-sampling factor $N$ and patch size $R$, modify the way in which we ternarize image patches, or analyze different image datasets (*Figure 6A* and Appendix 3). Eliminating downsampling completely (choosing $N = 1$) does lead to slightly larger mismatches between predicted and measured thresholds (first three distributions on the left in *Figure 6A*) as expected from *Hermundstad et al., 2014*, a finding that we attribute to artifacts arising from imperfect demosaicing of the camera's filter array output.

## Invariances in psychophysics recapitulate symmetries in natural images

The 'variance is salience' hypothesis can be further tested by asking whether symmetries of the natural distribution of textures are reflected in invariances of psychophysical thresholds. Binary texture coordinates (*Hermundstad et al., 2014*) are not affected by many of these symmetry transformations, and so a test requires textures containing at least three gray levels. For instance, reflecting a texture around the vertical axis has no effect on second-order statistics in the binary case, but it leads to a flip around the $p_0$ direction in the $\beta_{+-}$ simple plane in ternary texture space (dashed green line in *Figure 4C*). To see this, recall that the coordinates in the $\beta_{+-}$ plane are given by the probabilities that $\mathrm{mod}(A_1 - A_2, 3) = h$ for the three possible values of $h$ (*Figure 1*). Under a left-right flip, the values $A_1$ and $A_2$ are exchanged, leading to $\mathrm{mod}(A_1 - A_2, 3) = \mathrm{mod}(-h, 3)$. This means that the $h = 1$ direction gets mapped to $h = 2$, the $h = 2$ direction gets mapped to $h = 1$, and the $h = 0$ direction remains unaffected. More details, and a generalization to additional symmetry transformations, can be found in Appendix 3. We find that the distribution of natural images is symmetric about the $p_0$ direction and is thus unaffected by this transformation, predicting that psychophysical thresholds should also be unaffected when textures are flipped about the vertical axis. This is indeed the case (*Figure 6B*). Similarly, the natural image distribution is symmetric under flips about the horizontal axis, and also under rotations by 90, 180, and 270 degrees, predicting perceptual invariances that are borne out in the psychophysical data (*Figure 6B*).

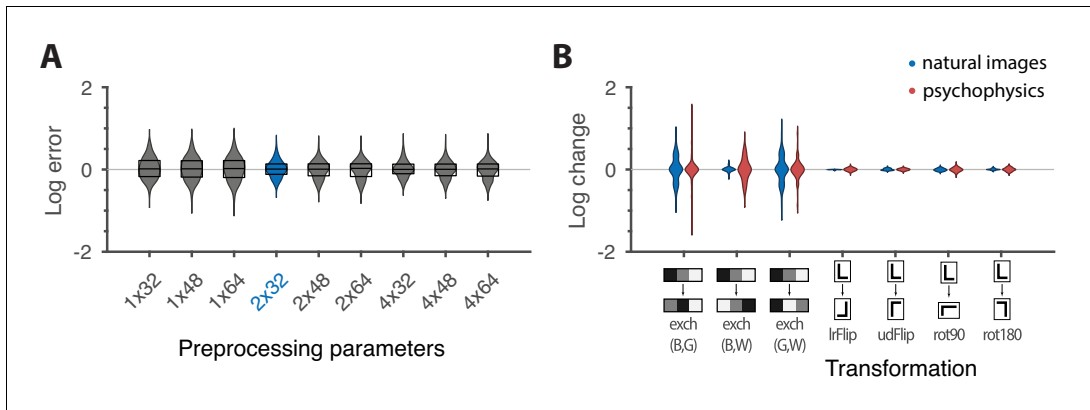

**Figure 6.** Robustness of results and effects of symmetry transformations. (**A**) The difference between the natural logarithms of the measured and predicted thresholds (red crosses and blue dots, respectively, in *Figures 4C* and *5*) is approximately independent of the downsampling ratio $N$ and patch size $R$ used in preprocessing. The labels on the x-axis are in the format $N \times R$, with the violin plot and label in blue representing the analysis that we focused on in the rest of the paper. Each violin plot in the figure shows a kernel density estimate for the distribution of prediction errors for the 311 second-order single- and mixed-plane threshold measurements available in the psychophysics. The boxes show the 25th and 75th percentiles, and the lines indicate the medians. (**B**) Change in the natural logarithms of predicted (blue) or measured (red) thresholds following a symmetry transformation. Symmetry transformations that leave the natural image predictions unchanged also leave the psychophysical measurements unchanged. (See text for the special case of the exch(B,W) transformation.) The visualization style is the same as in panel **A**, except boxes and medians are not shown. The transformations starting with exch correspond to exchanges between gray levels; e.g., exch(B,W) exchanges black and white. lrFlip and udFlip are left-right and up-down geometric flips, respectively, while rot90 and rot180 are geometric rotations by the respective number of degrees (clockwise).

Reflecting a texture about the vertical axis also has an interesting effect on the $\beta_{+-}$ plane: it not only flips the texture about the $p_0$ direction, but it also maps the texture onto the plane corresponding to the *opposite* diagonal orientation, $\beta_{-+}$. The fact that a flip about the $p_0$ direction is a symmetry of natural images is thus related to the fact that the diagonal pairwise correlations are the same regardless of the orientation of the diagonal. This fact was already observed in the binary analysis (*Hermundstad et al., 2014*), and is related to the invariance under 90-degree rotations observed here (*Figure 6B*).

It is important to note that these symmetries were not guaranteed to exist for either natural images or human psychophysics. Most of the textures that we are using are not themselves invariant under rotations (see the examples from *Figure 1C,D*). This means that invariances of predicted thresholds arise from symmetries in the overall shape of the *distribution* of natural textures. Similarly, had observed thresholds been unrelated to natural texture statistics, we could have found a discrepancy between the symmetries observed in natural images and those observed in human perception. As an example, the up and down directions differ in meaning, as do vertical and horizontal directions. A system that preserves these semantic differences would not be invariant under flips and rotations. The fact that the psychophysical thresholds are, in fact, invariant under precisely those transformations that leave the natural image distribution unchanged supports the idea that this is an adaptation to symmetries present in the natural visual world.

Natural images also have a well-known asymmetry between bright and dark contrasts (*Ratliff et al., 2010*; *Tkacik et al., 2010*) that is reflected in the anatomy and physiology of visual circuits, and in visual behavior (*Ratliff et al., 2010*; *Tkacik et al., 2010*; *Zemon et al., 1988*; *Chubb et al., 2004*; *Jin et al., 2008*; *Komban et al., 2014*; *Kremkow et al., 2014*). Our psychophysical data also show a bright/dark asymmetry. For instance, in *Figure 4C*, the threshold contour is not symmetric under the exchange of black and white checks, which has the effect of reflecting thresholds about the upper-left axis in the $\beta_{++}$ plane (dashed orange line in the figure). Such bright-dark asymmetries lead to a wide distribution of relative changes in detection threshold upon the exchange of black and white (red violin plot in *Figure 6B* for the exch(B,W) transformation). Our

natural image analysis does not show this asymmetry (blue violin plot in *Figure 6B* for the exch(B,W) transformation) because of our preprocessing, which follows previous work (*Hermundstad et al., 2014*). As observed in *Ratliff et al., 2010*, the bright-dark asymmetry rests on two main characteristics of natural images: a skewed distribution of light intensities such that the mean intensity is larger than the median, and a power-law spectrum of spatial correlations. Both of these are reduced or removed by our preprocessing pipeline, which starts by taking the logarithm of intensity values and thereby reduces the skewness in the intensity distribution, and continues with a whitening stage that removes the overall $1/f$ spatial correlation spectrum seen in natural images. The final ternarization step additionally reduces any remaining dark-bright asymmetry, since we ensure that each of the three gray levels occurs in equal proportions in the preprocessed patches. This explains why we do not see this asymmetry in our natural-image analysis.

## Discussion

The efficient coding hypothesis posits that sensory systems are adapted to maximize information about natural sensory stimuli. In this work, we provided a rigorous quantitative test of this hypothesis in the context of visual processing of textures in a regime dominated by sampling noise. To this end, we extended the study of binary texture perception to grayscale images that capture a broader range of correlations to which the brain could conceivably adapt. We first generalized the definition of textures based on local multi-point correlations to accommodate multiple luminance levels. We then constructed algorithms for generating these textures, and we used these in our behavioral studies. By separately analyzing the distribution of textures across an ensemble of natural images, we showed that psychophysical thresholds can be predicted in remarkable detail based on the statistics of natural scenes. By further exploiting symmetry transformations that have non-trivial effects on ternary (but not binary) texture statistics, we provided a novel test of efficient coding and therein demonstrated that visually-guided behavior shows the same invariances as the distribution of natural textures. Overall, this work strengthens and refines the idea that the brain is adapted to efficiently encode visual texture information.

The methodology developed here can be used to address many hypotheses about visual perception. For example, if a specific set of textures was hypothesized to be particularly ethologically relevant, this set could be measured and compared against 'irrelevant' textures of equal signal-to-noise ratio. Because our hypothesis treats every dimension of texture space equally—the symmetry only broken by properties of the natural environment—we leveraged the rapidly increasing dimensionality of grayscale texture space to more stringently test the efficient coding hypothesis. In this vein, our construction can be generalized to larger numbers of gray levels and correlations over greater distances. However, the ability of neural systems to adapt to such correlations must ultimately be limited because, as texture complexity grows, it will eventually become impossible for the brain to collect sufficient statistics to determine optimal sensitivities. Even were it possible to accumulate these statistics, adapting to them might not be worth the computational cost of detecting and processing long-range correlations between many intensity values. Understanding the limits of texture adaptation will teach us about the cost-benefit tradeoffs of efficient coding in sensory cortex, in analogy with recently identified cost-benefit tradeoffs in optimal inference (*Tavoni et al., 2019*). And indeed, although our predictions are in excellent agreement with the data in most cases, we find a few systematic differences that may already be giving us a glimpse of these limits.

## Materials and methods

### Code and data

The code and data used to generate all of the results in the paper can be found on GitHub (RRID: SCR_002630), at https://github.com/ttesileanu/TextureAnalysis (*Tesileanu et al., 2020*; copy archived at https://github.com/elifesciences-publications/TextureAnalysis).

### Definition of texture space

A texture is defined here by the statistical properties of $2 \times 2$ blocks of checks, each of which takes the value 0, 1, or 2, corresponding to the three luminance levels (black, gray, or white; see Appendix

1 for a generalization to more gray levels). The $3^4 = 81$ probabilities for all the possible configurations of such blocks form an overcomplete coordinate system because the statistical properties of textures are independent of position. To build a non-redundant parametrization of texture space, we use a construction based on a discrete Fourier transform (see Appendix 1). Starting with the luminance values $A_i$, $i = 1, \ldots, 4$, of checks in a $2 \times 2$ texture block (arranged as in **Figure 1A**), we define the coordinates $\sigma_{\substack{s_1 s_2 \\ s_3 s_4}}(h)$ which are equal to the fraction of locations where the linear combination $s_1 A_1 + s_2 A_2 + s_3 A_3 + s_4 A_4$ has remainder equal to $h$ after division by three (the number of gray levels). In the case of three gray levels, the coefficients $s_i$ can be +1, $-1$, or 0.

Each set of coefficients $s_i$ identifies a texture group, and within each texture group we have three probability values, one for each value of $h$. Since the probabilities sum up to 1, each texture group can be represented as a plane, and more specifically, as a triangle in a plane, since the probabilities are also non-negative. This is the representation shown in **Figure 1C,D** and used in subsequent figures. For compactness of notation, when referring to the coefficients $s_i$, we write + and $-$ instead of +1 and $-1$, and omit coefficients that are 0, e.g., $\sigma_{\substack{+ - \\ -}}$ instead of $\sigma_{\substack{+1 -1 \\ 0 -1}}$. We also use $\gamma$ (rather than the generic symbol $\sigma$) for 1-point correlations, $\beta$ for 2-point correlations, $\theta$ for 3-point correlations, and $\alpha$ for 4-point correlation, matching the notation used in the binary case (**Victor and Conte, 2012**; **Hermundstad et al., 2014**). For instance, $\beta_{\substack{+ \\ -}}$ is the plane identified by the linear combination $A_1 - A_3 \pmod 3$.

## Texture analysis and synthesis

Finding the location in texture space that matches the statistics of a given image patch is straightforward given the definition above: we simply glide a $2 \times 2$ block over the image patch and count the fraction of locations where the combination $s_1 A_1 + s_2 A_2 + s_3 A_3 + s_4 A_4$ takes each of its possible values modulo three, for each texture group identified by the coefficients $s_i$. (As a technical detail, we glide the smallest shape that contains non-zero coefficients. For example, for $\beta_{+-}$, we glide a $1 \times 2$ region instead of a $2 \times 2$ one. This differs from gliding the $2 \times 2$ block for all orders only through edge effects, and thus the difference decreases as the patch size $R$ increases.)

In order to generate a patch that corresponds to a given location in texture space, we use a maximum entropy construction that is an extension of the methods from **Victor and Conte, 2012**. There are efficient algorithms based on 2d Markov models that can generate all single-group textures, namely, textures in which only the probabilities within a single texture group deviate from (1/3, 1/3, 1/3). For textures involving several groups, the construction is more involved, and in fact some combinations of texture coordinates cannot be achieved in a real texture. This restriction applies as textures become progressively more 'saturated', that is, near the boundary of the space. In contrast, near the origin of the space all combinations can be achieved via the 'donut' construction for mixing textures, described in **Victor and Conte, 2012**. Details are provided in Appendix 2.

## Visual stimulus design

The psychophysical task is adapted from **Victor et al., 2005**, and requires that the subject identify the location of a $16 \times 64$-check target within a $64 \times 64$-check array. The target is positioned near one of the four sides of the square array (chosen at random), with an 8-check margin. Target and background are distinguished by the texture used to color the checks: one is always the *i.i.d.* (unbiased) texture with three gray levels; the other is a texture specified by one or two of the coordinates defined in the text. In half of the trials, the target is structured and the background is *i.i.d.*; in the other half of the trials, the target is *i.i.d.* and the background is structured. To determine psychophysical sensitivity in a specific direction in the space of image statistics, we proceed as follows **Hermundstad et al., 2014**; **Victor and Conte, 2012**; **Victor et al., 2013**; **Victor et al., 2015**. We measure subject performance in this 4-alternative forced-choice task across a range of 'texture contrasts', that is, distances from the origin in the direction of interest. Fraction correct, as a function of texture contrast, is fit to a Weibull function, and threshold is taken as the texture contrast corresponding to a fraction correct of 0.625, that is, halfway between chance (0.25) and ceiling (1.0). Typically, 12 different directions in one plane of stimulus space are studied in a randomly interleaved fashion. Each of these 12 directions is sampled at 3 values of texture contrast, chosen in pilot experiments to yield performance between chance and ceiling. These trials are organized into 15 blocks of 288 trials each (a total of 4320 trials), so that each direction is sampled 360 times.

### Visual stimulus display

Stimuli, as described above, were presented on a mean-gray background for 120 ms, followed by a mask consisting of an array of *i.i.d.* checks, each half the size of the stimulus checks. The display size was $15 \times 15$ deg; viewing distance was 103 cm. Each of the $64 \times 64$ array stimulus checks consisted of $10 \times 10$ hardware pixels, and measured $14 \times 14$ min. The display device was an LCD monitor with a refresh rate of 100 Hz, driven by a Cambridge Research ViSaGe system. The monitor was calibrated with a photometer prior to each day of data collection to ensure that the luminance of the gray checks was halfway between that of the black checks (<0.1) and white checks (23 cd/m$^2$).

### Psychophysics subjects

Subjects were normal volunteers (three male, three female), ages 20 to 57, with visual acuities, corrected if necessary, of 20/20 or better. Of the six subjects, MC is an experienced psychophysical observer with thousands of hours of experience; the other subjects (SR, NM, WC, ZA, JWB) had approximately 10 (JWB), 40 (NM, WC, ZA) or 100 (SR) hours of experience at the start of the study, as subjects in related experiments. MC is an author. NM, WC, and ZA were naïve to the purposes of the experiment.

This work was carried out with the subjects' informed consent, and in accordance with the Code of Ethics of the World Medical Association (Declaration of Helsinki) and the approval of the Institutional Review Board of Weill Cornell.

### Psychophysics averaging

The average thresholds used in the main text were calculated by using the geometric mean of the subject thresholds, after applying a per-subject scaling factor chosen to best align the overall sensitivities of all the subjects; these multipliers ranged from 0.855 to 1.15. Rescaling a single consensus set of thresholds in this fashion accounted for 98.8% of the variance of individual thresholds across subjects. The average error bars were calculated by taking the root-mean-squared of the per-subject error bars in log space (determined from a bootstrap resampling of the Weibull-function fits, as in *Victor et al., 2005*), and then exponentiating.

### Natural image preprocessing

Images were taken from the UPenn Natural Image Database (*Tkačik et al., 2011*) and preprocessed as shown in *Figure 3A* (see Materials and methods for details). Starting with a logarithmic encoding of luminance, we downsampled each image by averaging over $N \times N$ blocks of pixels to reduce potential camera sampling artifacts. We then split the images into non-overlapping patches of size $R$, filtered the patches to remove the average pairwise correlation expected in natural images (*van Hateren, 1992*), and finally ternarized patches to produce equal numbers of black, gray, and white checks (*Figures 3A, C and D*). For most figures shown in the main text, we used $N = 2$ and $R = 32$. Each patch was then analyzed in terms of its texture content and mapped to a point in ternary texture space following the procedure described in the main text. Finally, to avoid biases due to blurring artifacts, we fit a two-Gaussian mixture model to the texture distribution and used this to separate in-focus from blurred patches (*Figure 3B*; *Hermundstad et al., 2014*; details in Materials and methods).

### Whitening of natural image patches

To generate the whitening filter that we used to remove average pairwise correlations, we started with the same preprocessing steps as for the texture analysis, up to and including the splitting into non-overlapping patches (first three steps in *Figure 3A*). We then took the average over all the patches of the power spectrum, which was obtained by taking the magnitude-squared of the 2d Fourier transform. Taking the reciprocal square root of each value in the resulting matrix yielded the Fourier transform of the filtering matrix.

### Removal of blurred patches in natural images

Following the procedure from *Hermundstad et al., 2014*, we fit a Gaussian mixture model with non-shared covariance matrices to the distribution of natural images in order to identify patches that are out of focus or motion blurred. This assigned each image patch to one of two multivariate Gaussian

distributions. To identify which mixture component contained the sharper patches, we chose the component that had the higher median value of a measure of sharpness based on a Laplacian filter. Specifically, each patch was normalized so that its median luminance was set to 1, then convolved with the matrix $\begin{pmatrix} 0 & 1 & 0 \\ 1 & -4 & 1 \\ 0 & 1 & 0 \end{pmatrix}$. The sharpness of the patch was calculated as the median absolute value over the pixels of the convolution result. This analysis was performed before any of the preprocessing steps, including the transformation to log intensities, and was restricted to the pixels that did not border image edges. There was thus no need to make assumptions regarding pixel values outside images.

## Efficient coding calculations

Threshold predictions from natural image statistics were obtained as in *Hermundstad et al., 2014*. We fit a multivariate Gaussian to the distribution of texture patches after removing the blurry component, and used the inverse of the standard deviation in the texture direction of interest as a prediction for the psychophysical threshold in that direction. The overall scale of the predictions is not fixed by this procedure. We chose the scaling so as to minimize the error between the $n$ measurements $x_i$ and the predictions $y_i$, $\min \frac{1}{n} \sum_{i=1}^{n} (\frac{\log y_i - \log x_i}{\epsilon_i})^2$, where $\epsilon_i$ are the measurement uncertainties in log space. This scaling factor was our single fitting parameter.

## Calculating mismatch

In *Figure 6* we show several comparisons of two sets of thresholds $x_i$ and $y_i$. These are either the set of measured thresholds and the set of natural image predictions for specific preprocessing options; or sets of either measured or predicted thresholds before and after the action of a symmetry transformation. We measure mismatch by the difference between the natural logarithms of the two quantities, $\log y_i - \log x_i$, which is approximately equal to the relative error when the mismatches are not too large ($\log y_i - \log x_i = \log y_i / x_i = \log(1 + \frac{y_i - x_i}{x_i}) \approx \frac{y_i - x_i}{x_i}$). In panel A of the figure all 311 measured values and the corresponding predictions were used. For panel B, this set was restricted in two ways. First, for we ignored the measurements for which we did not have psychophysics data for the transformed direction. And second, we ignored directions on which the transformation acted trivially (see Appendix 3).

## Acknowledgements

VB was supported by the US–Israel Binational Science Foundation grant 2011058 and by the National Science Foundation Physics Frontiers Center Grant PHY-1734030. VB, JDV, and MC were supported by EY07977. TT was supported by the Swartz Foundation during part of this work. JJB and AMH were supported by the Howard Hughes Medical Institute. A portion of this work was presented at the Society for Neuroscience (2018) and Vision Sciences Society (2017, 2018).

## Additional information

### Funding

| Funder | Grant reference number | Author |
| --- | --- | --- |
| United States-Israel Binational Science Foundation | 2011058 | Vijay Balasubramanian |
| National Eye Institute | EY07977 | Mary M Conte<br>Jonathan D Victor<br>Vijay Balasubramanian |
| The Swartz Foundation | | Tiberiu Tesileanu |
| Howard Hughes Medical Institute | | John J Briguglio<br>Ann M Hermundstad |

| National Science Foundation | Physics Frontiers Center PHY-1734030 | Vijay Balasubramanian |

The funders had no role in study design, data collection and interpretation, or the decision to submit the work for publication.

### Author contributions

Tiberiu Tesileanu, Conceptualization, Software, Formal analysis, Validation, Visualization, Methodology, Writing - original draft, Writing - review and editing; Mary M Conte, Conceptualization, Resources, Data curation, Investigation, Methodology; John J Briguglio, Ann M Hermundstad, Software, Formal analysis, Writing - review and editing; Jonathan D Victor, Conceptualization, Resources, Data curation, Software, Supervision, Funding acquisition, Methodology, Project administration, Writing - review and editing; Vijay Balasubramanian, Conceptualization, Supervision, Funding acquisition, Methodology, Project administration, Writing - review and editing

### Author ORCIDs

Tiberiu Tesileanu (iD) https://orcid.org/0000-0003-3107-3088
Ann M Hermundstad (iD) http://orcid.org/0000-0002-0377-0516
Jonathan D Victor (iD) https://orcid.org/0000-0002-9293-0111
Vijay Balasubramanian (iD) http://orcid.org/0000-0002-6497-3819

### Ethics

Human subjects: This work was carried out with the subjects' informed consent, and in accordance with the Code of Ethics of the World Medical Association (Declaration of Helsinki) and the approval of the Institutional Review Board of Weill Cornell. The IRB protocol number is 0904010359.

### Decision letter and Author response

Decision letter https://doi.org/10.7554/eLife.54347.sa1
Author response https://doi.org/10.7554/eLife.54347.sa2

## Additional files

### Supplementary files

• Transparent reporting form

### Data availability

All the code and data necessary to reproduce the results from the manuscript are available at https://github.com/ttesileanu/TextureAnalysis (copy archived at https://github.com/elifesciences-publications/TextureAnalysis).

The following datasets were generated:

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

# Appendix 1

## A coordinate system for local image statistics

This Supplement provides the details for the present parameterization of local image statistics. It generalizes the approach developed in *Victor and Conte, 2012* for binary images along the lines indicated in that manuscript's Appendix A, so that it is applicable to an arbitrary number $G$ of gray levels. In the present study, $G = 3$, but the analysis is equally applicable to any prime value $G$, including the $G = 2$ case that has been the focus of previous work (*Briguglio et al., 2013*; *Victor and Conte, 2012*; *Victor et al., 2017*; *Victor et al., 2013*; *Victor et al., 2015*). When $G$ is composite, the basic approach remains valid but, as mentioned below, there are some additional considerations.

We consider the statistics of a $2 \times 2$ neighborhood of checks in images with $G$ gray levels. The starting point is an enumeration of the probabilities of each kind of $2 \times 2$ block, $p \begin{pmatrix} A_1 & A_2 \\ A_3 & A_4 \end{pmatrix}$, where each $A_k$ denotes the gray level of a check, which we denote by an integer from 0 to $G - 1$. There are $G^4$ such configurations, but the probabilities are not independent: they must sum to 1, and they must be stationary in space. For example, the probability of $1 \times 2$ blocks computed by marginalizing over the lower two checks must equal to the probability of $1 \times 2$ blocks computed by marginalizing over the upper two checks.

Our main goal is to obtain a coordinate system that removes these linear dependencies. The first step is to construct new coordinates $\varphi \begin{pmatrix} s_1 & s_2 \\ s_3 & s_4 \end{pmatrix}$ by discrete Fourier transformation with respect to the gray level value in each check. As this is a discrete transform, the arguments $s_k$ are also integers from 0 to $G - 1$. Equivalently, we can use other sets of integers that lead to unique values of the complex exponential; for instance, when $G = 3$, it is equivalent to use the set $\{0, +1, -1\}$ or the set $\{0, 1, 2\}$. We have

$$\varphi \begin{pmatrix} s_1 & s_2 \\ s_3 & s_4 \end{pmatrix} = \sum_{A_1=0}^{G-1} \sum_{A_2=0}^{G-1} \sum_{A_3=0}^{G-1} \sum_{A_4=0}^{G-1} p \begin{pmatrix} A_1 & A_2 \\ A_3 & A_4 \end{pmatrix} e^{-\left(\frac{2\pi i}{G}\right)(A_1 s_1 + A_2 s_2 + A_3 s_3 + A_4 s_4)} \, . \tag{1}$$

The original block probabilities $p \begin{pmatrix} A_1 & A_2 \\ A_3 & A_4 \end{pmatrix}$ can be obtained from the Fourier transform coordinates (*Equation (1)*) by standard inversion:

$$p \begin{pmatrix} A_1 & A_2 \\ A_3 & A_4 \end{pmatrix} = \frac{1}{G^4} \sum_{s_1=0}^{G-1} \sum_{s_2=0}^{G-1} \sum_{s_3=0}^{G-1} \sum_{s_4=0}^{G-1} \varphi \begin{pmatrix} s_1 & s_2 \\ s_3 & s_4 \end{pmatrix} e^{\left(\frac{2\pi i}{G}\right)(A_1 s_1 + A_2 s_2 + A_3 s_3 + A_4 s_4)} \, . \tag{2}$$

Fourier transform coordinates can be similarly described for any configuration of checks, including subsets of the $2 \times 2$ neighborhood.

A basic property of the Fourier transform coordinates is that setting an argument to zero corresponds to marginalizing over the corresponding check. For example, consider the probabilities of the configurations of the upper $1 \times 2$ block of checks, $p \begin{pmatrix} A_1 & A_2 \end{pmatrix}$, which are determined by marginalizing $p \begin{pmatrix} A_1 & A_2 \\ A_3 & A_4 \end{pmatrix}$ over $A_3$ and $A_4$. Their Fourier transform coordinates are given by

$$\varphi \begin{pmatrix} s_1 & s_2 \end{pmatrix} = \sum_{A_1=0}^{G-1} \sum_{A_2=0}^{G-1} p \begin{pmatrix} A_1 & A_2 \end{pmatrix} e^{-\left(\frac{2\pi i}{G}\right)(A_1 s_1 + A_2 s_2)} \, . \tag{3}$$

It follows from *Equation (1)* that

$$\varphi\begin{pmatrix} s_1 & s_2 \end{pmatrix} = \sum_{A_1=0}^{G-1}\sum_{A_2=0}^{G-1} p\begin{pmatrix} A_1 & A_2 \end{pmatrix} e^{-(\frac{2\pi i}{G})(A_1 s_1 + A_2 s_2)}$$

$$= \sum_{A_1=0}^{G-1}\sum_{A_2=0}^{G-1}\left[\sum_{A_3=0}^{G-1}\sum_{A_4=0}^{G-1} p\begin{pmatrix} A_1 & A_2 \\ A_3 & A_4 \end{pmatrix}\right] e^{-(\frac{2\pi i}{G})(A_1 s_1 + A_2 s_2)} \tag{4}$$

$$= \varphi\begin{pmatrix} s_1 & s_2 \\ 0 & 0 \end{pmatrix}.$$

Thus, the stationarity condition

$$p\begin{pmatrix} A_1 & A_2 \end{pmatrix} = p\begin{pmatrix} \\ A_1 & A_2 \end{pmatrix} \tag{5}$$

is equivalent to

$$\varphi\begin{pmatrix} s_1 & s_2 \\ 0 & 0 \end{pmatrix} = \varphi\begin{pmatrix} 0 & 0 \\ s_1 & s_2 \end{pmatrix}. \tag{6}$$

Similarly, the stationarity condition for $2 \times 1$ blocks is equivalent to

$$\varphi\begin{pmatrix} s_1 & 0 \\ s_3 & 0 \end{pmatrix} = \varphi\begin{pmatrix} 0 & s_1 \\ 0 & s_3 \end{pmatrix}; \tag{7}$$

the condition that the single-check probabilities are equal in all four positions is equivalent to

$$\varphi\begin{pmatrix} s & 0 \\ 0 & 0 \end{pmatrix} = \varphi\begin{pmatrix} 0 & s \\ 0 & 0 \end{pmatrix} = \varphi\begin{pmatrix} 0 & 0 \\ s & 0 \end{pmatrix} = \varphi\begin{pmatrix} 0 & 0 \\ 0 & s \end{pmatrix}; \tag{8}$$

and the condition that the sum of all block probabilities is one is equivalent to

$$\varphi\begin{pmatrix} 0 & 0 \\ 0 & 0 \end{pmatrix} = 1. \tag{9}$$

In sum, the stationarity conditions can be stated in terms of the Fourier transform coordinates as follows: if any of the arguments of $\varphi\begin{pmatrix} s_1 & s_2 \\ s_3 & s_4 \end{pmatrix}$ are zero, then they can be replaced by empty spaces, and the value of $\varphi\begin{pmatrix} s_1 & s_2 \\ s_3 & s_4 \end{pmatrix}$ must be unchanged by translating the nonzero values within the $2 \times 2$ neighborhood. It follows that the Fourier transform coordinates of a stationary distribution are specified by: $\varphi(s)$, equal to the common value of the four expressions in *Equation (8)*; $\varphi(s_1 \ s_2)$, equal to the common value of the two expressions in *Equation (6)*; $\varphi\begin{pmatrix} s_1 \\ s_3 \end{pmatrix}$, equal to the common value of the two expressions in *Equation (7)*; $\varphi\begin{pmatrix} s_1 & \\ & s_4 \end{pmatrix}$ and $\varphi\begin{pmatrix} & s_2 \\ s_3 & \end{pmatrix}$, defining pairwise correlations of two-check configurations that cannot be translated within the $2 \times 2$ neighborhood; $\varphi\begin{pmatrix} & s_2 \\ s_3 & s_4 \end{pmatrix}$, $\varphi\begin{pmatrix} s_1 & \\ s_3 & s_4 \end{pmatrix}$, $\varphi\begin{pmatrix} s_1 & s_2 \\ & s_4 \end{pmatrix}$, and $\varphi\begin{pmatrix} s_1 & s_2 \\ s_3 & \end{pmatrix}$, defining three-check correlations, and $\varphi\begin{pmatrix} s_1 & s_2 \\ s_3 & s_4 \end{pmatrix}$. In all of these cases, the arguments $s_k$ are nonzero. Thus, the total number of parameters, obtained by allowing each of the $s_k$ to range from 1 to $G-1$, is $(G-1) + 4(G-1)^2 + 4(G-1)^3 + (G-1)^4 = G(G-1)(G^2 + G - 1)$; this is 10 for $G = 2$ and 66 for $G = 3$.

The Fourier transform coordinates incorporate the stationarity constraints, and they also have the convenient property that the origin of the space, that is, the image whose coordinates are all zero, is an image with identically-distributed gray levels in which every block probability $p\begin{pmatrix} A_1 & A_2 \\ A_3 & A_4 \end{pmatrix}$ is equal to $1/G^4$. The Fourier transform coordinates have another convenient property (using the approach of Appendix B of *Victor and Conte, 2012*): near the origin of the space, entropy is asymptotically proportional to the square of the Euclidean distance from the origin. This means that the Fourier

transform coordinates are 'calibrated': for an ideal observer, small deviations in any direction from the origin are equally discriminable.

However, since for $G > 2$, the Fourier transform coordinates are complex numbers, an arbitrary choice of them will typically not correspond to a realizable set of block probabilities, since the block probabilities must be all real and in the range [0, 1].

To address this problem, we note a relationship among the Fourier transform coordinates, which, when $G$ is prime, partitions the set of independent coordinates into disjoint subsets of size $G - 1$. A further linear transformation within this set yields real-valued coordinates, which correspond to the coordinate system used here (for $G = 3$) and in previous studies (for $G = 2$).

To derive these real-valued coordinates, we use modular arithmetic. When $G$ is prime, for every integer $q \in \{1, \ldots, G - 1\}$ there is a unique integer $r \in \{1, \ldots, G - 1\}$ for which $qr = 1 \pmod{G}$, which we denote $q^{-1}$. Thus, if $S' = \begin{pmatrix} s_1' & s_2' \\ s_3' & s_4' \end{pmatrix}$ can be obtained from $S = \begin{pmatrix} s_1 & s_2 \\ s_3 & s_4 \end{pmatrix}$ by $S' = qS$, then it follows that $S$ can be obtained from $S'$ by $S = q^{-1}S'$, that is, the relationship is reciprocal. It also follows that every Fourier transform coordinate $S'$ is some scalar multiple of a 'monic' Fourier transform coordinate, that is, one whose first nonzero element is 1. The reason is that we can take the multiplier $q$ to be the first nonzero coordinate of $S'$ and write $S = q^{-1}S'$.

Next, we note that all of the Fourier transform coordinates whose arguments are of the form $qS$ have an exponential in *Equation (1)* that depends only on the value of $\sum_{k=1}^{4} s_k A_k \pmod{G}$. This motivates grouping the terms of *Equation (1)* according to this value. We therefore define

$$\sigma_{\begin{pmatrix} s_1 & s_2 \\ s_3 & s_4 \end{pmatrix}}(h) = \sum_{A_1=0}^{G-1} \sum_{A_2=0}^{G-1} \sum_{A_3=0}^{G-1} \sum_{A_4=0}^{G-1} p\begin{pmatrix} A_1 & A_2 \\ A_3 & A_4 \end{pmatrix} \delta_{\mathrm{mod}\,G}(A_1 s_1 + A_2 s_2 + A_3 s_3 + A_4 s_4 - h), \qquad (10)$$

that is, $\sigma_{\begin{pmatrix} s_1 & s_2 \\ s_3 & s_4 \end{pmatrix}}(h)$ is the sum of the probabilities of all blocks for which $\sum_{k=1}^{4} s_k A_k = h \pmod{G}$. The monic $\sigma$'s, that is, the $\sigma$'s whose first nonzero $s_k$ is equal to 1, are our desired real-valued coordinates. For each such $\sigma$, the coordinates are a vector of the $G$ numbers $\sigma_{\begin{pmatrix} s_1 & s_2 \\ s_3 & s_4 \end{pmatrix}}(h)$, for $h = \{0, \ldots, G - 1\}$. As *Equation (10)* shows, $\sigma_{\begin{pmatrix} s_1 & s_2 \\ s_3 & s_4 \end{pmatrix}}(h)$ is the probability that a linear combination of gray levels whose coefficients are specified by the $s_k$ will result in a value of $h$. They are thus all in the range [0, 1], and their sum is 1. For $G = 2$, the pair $\sigma(0)$ and $\sigma(1)$ subject to $\sigma(0) + \sigma(1) = 1$ is a one-dimensional domain, parametrized by $\sigma(1) - \sigma(0)$; other than a change in sign, these are the coordinates used in *Briguglio et al., 2013*; *Victor and Conte, 2012*; *Victor et al., 2017*; *Victor et al., 2013*; *Victor et al., 2015*. For $G = 3$, as is the case here, the triplet $\sigma(0)$, $\sigma(1)$, and $\sigma(2)$ is subject to $\sigma(0) + \sigma(1) + \sigma(2) = 1$ and is naturally plotted in a triangular 'alloy plot' with centroid at (1/3, 1/3, 1/3), the image with independent, identically-distributed gray levels. Note that $2 = -1 \pmod 3$, yielding the notation in the main text where the coefficients $s_i$ took values 0, +1, and −1 instead of 0, 1, and 2.

It remains to show that all of the degrees of freedom in the Fourier Transform coordinates $\varphi$ are captured by the $\sigma$'s. Since the Fourier transform coordinates are partitioned into disjoint subsets, it suffices to examine the transformation between each of these subsets (i.e., between the Fourier transform coordinates that are scalar multiples of a particular monic $S$, and the corresponding $\sigma$). These subsets correspond to the simple planes introduced in the main text. It follows from *Equations (1) and (10)* that, for $q \neq 0 \pmod{G}$,

$$\varphi\begin{pmatrix} qs_1 & qs_2 \\ qs_3 & qs_4 \end{pmatrix} = \sum_{h=0}^{G-1} \sigma_{\begin{pmatrix} s_1 & s_2 \\ s_3 & s_4 \end{pmatrix}}(h) e^{-\left(\frac{2\pi i}{G}\right)qh}. \qquad (11)$$

Via discrete Fourier inversion, this equation implies

$$\sigma_{\begin{pmatrix} s_1 & s_2 \\ s_3 & s_4 \end{pmatrix}}(h) = \frac{1}{G}\sum_{q=0}^{G-1} \varphi\begin{pmatrix} qs_1 & qs_2 \\ qs_3 & qs_4 \end{pmatrix} e^{(\frac{2\pi i}{G})qh}. \tag{12}$$

Note that the $q=0$ term of this equation is given by *Equation (9)*, the normalization condition. Thus,

$$\sigma_{\begin{pmatrix} s_1 & s_2 \\ s_3 & s_4 \end{pmatrix}}(h) = \frac{1}{G} + \frac{1}{G}\sum_{q=1}^{G-1} \varphi\begin{pmatrix} qs_1 & qs_2 \\ qs_3 & qs_4 \end{pmatrix} e^{(\frac{2\pi i}{G})qh}. \tag{13}$$

*Equations (11) and (13)* display the bidirectional transformation between the $G$-vector $\sigma$ and the $G-1$ Fourier transform coordinates $qS$. Since this transformation is a discrete Fourier transform (and hence, a multiple of a unitary transformation), it preserves the property that entropy is a proportional to the square of the Euclidean distance from the origin.

When $G$ is not prime, the decomposition of Fourier transform coordinates into disjoint sets indexed by monic coordinates is no longer possible. For example, with $G=4$, both $S' = \begin{pmatrix} 1 & 0 \\ 0 & 0 \end{pmatrix}$ and $S'' = \begin{pmatrix} 1 & 2 \\ 2 & 0 \end{pmatrix}$ are monic but the sets $qS'$ and $qS''$ have a non-empty intersection: $2S' = 2S'' = \begin{pmatrix} 2 & 0 \\ 0 & 0 \end{pmatrix} \pmod 4$. This necessitates a more elaborate version of the above approach, with separate strata for each factor of $G$. The disjoint sets of Fourier transform coordinates are no longer all of the same size, and the transformations (*Equation 11*) and (*Equation 13*) have a more elaborate form, although the overall parameter count is the same. As this case is not relevant to the present experiments, we do not discuss it further.

## Appendix 2

### Texture synthesis

Creation of the stimuli used here requires synthesis of images whose statistics are specified by one or two of the coordinates described above. We adopted the strategy of *Victor and Conte, 2012*, Table 2, for this purpose. (i) Coordinates that are of lower order than the specified coordinates are set to zero. The rationale is that sensitivity to lower-order coordinates is generally greater than sensitivity to higher-order coordinates, so this allows detection to be driven by the higher-order coordinates. (ii) Coordinates that are of higher order than the specified coordinates are chosen to achieve an image ensemble whose entropy is maximized. The rationale is that this sets the higher-order coordinates at the value that is implied by the lower-order coordinates, without any further spatial structure. (iii) Coordinates that are of the same order as the specified coordinates are chosen to satisfy the 'Pickard rules' (*Pickard, 1980*), as this allows for stimulus generation via a Markov process in the plane, and hence, maximizes entropy given the constraints of the specified textures. (iv) In a few cases (all involving third-order statistics), the specified parameters are inconsistent with the Pickard rules, and in these cases, specific choices of texture parameters are made to allow for convenient texture synthesis. In all cases in which the unspecified texture parameters are assigned a nonzero value, this nonzero value decreases to zero rapidly (at least quadratically) near the origin. Thus, the construction guarantees that the surface corresponding to any two specified texture coordinates is tangent to their coordinate plane at the origin.

In general, the above strategy also applies for $G \geq 3$. We focus on specifying pairs of second-order coordinates, the stimuli used in this study. When the specified coordinates share the same linear combination (e.g., $\beta_{12}[0]$ and $\beta_{12}[1]$) or when the two specified coordinates involve different linear combinations but the same checks (e.g., $\beta_{1_1}[0]$ and $\beta_{1_2}[0]$), images may be synthesized by a Markov process that operates in the direction of the coupled checks. For example, if $\beta_{1s_2}$ is specified, a Markov process is used to create each row. The Markov generation procedure guarantees that the image is a sample from a maximum-entropy ensemble consistent with the specified coordinates. It implicitly defines the other texture coordinates in a manner consistent with the above policies established for $G = 2$.

For combinations of coordinates involving non-identical pairs of checks (e.g., $\beta_{1s_2}$ and $\beta_{1_{s_3}}$), a new construction is needed in some cases. The basic issue is that the Markov process that generates the correlations along rows may be inconsistent with the Markov process that generates the columns. To see how this can happen, consider $\beta_{11}[0] = 1$ and $\beta_{1_2}[1] = 1$. Taking into account stationarity, $\beta_{11}[0] = 1$ means that $A_1 + A_2 = A_3 + A_4 = 0$; $\beta_{1_2}[1] = 1$ means that $A_1 + 2A_3 = A_2 + 2A_4 = 1$. The inconsistency arises because the first set of equalities implies that $A_2 = -A_1$ and $A_4 = -A_3$, so $A_1 + 2A_3 = -(A_2 + 2A_4)$, which is inconsistent with the second set of equalities. The inconsistency can also be seen from an algebraic viewpoint. The horizontal correlation $\beta_{11}$ corresponds to a left-to-right Markov process that biases check pairs whose gray levels satisfy a multiplicative recurrence, $x_{n+1} = -x_n = 2x_n$. The vertical correlation $\beta_{1_2}$ corresponds to a top-to-bottom Markov process that biases check pairs whose gray levels satisfy an additive recurrence, $x_{n+1} = x_n - 1$. The inconsistency arises because these transformations do not commute.

To handle these cases, we use the following construction. It produces samples from an image ensemble that match the specified second-order statistics, and for which all other second-order statistics, as well as first- and third-order statistics, are zero. As a first step, two textures are created: a Markov process along rows, providing for correlations of the form $\beta_{1s_2}$, and a Markov process along columns, providing for correlations of the form $\beta_{1_{s_3}}$. Then, taking inspiration from the 'dead leaves' generative model of images (*Zylberberg et al., 2012*), we randomly choose rows from the first texture and columns from the second texture and sequentially place them on the plane, occluding whatever has been placed earlier. Eventually, this 'falling sticks' construction covers the plane, and this completes the next stage of the construction.

The resulting texture has the same kinds of correlations as the two starting textures, but the correlations are diluted, because a 'stick' in one direction may be overlaid by an orthogonal stick placed at a later time. To calculate this dilution, we consider the three 'sticks' that could contribute to a

given $1 \times 2$ block of the final texture: one horizontal stick and two vertical sticks. Since they are dropped in a random order, there is a 1/3 chance that any of them is the one that is placed last. If the last stick is horizontal (which happens with probability 1/3), then the two checks of the $1 \times 2$ block reflect the correlation structure of the underlying texture. If the last stick is vertical (which happens with probability 2/3), then these two checks are uncorrelated, since they are derived from independent Markov processes. That is,

$$
\begin{aligned}
\sigma^{\text{falling sticks}}\begin{pmatrix} s_1 & s_2 \end{pmatrix} &= \frac{1}{3}\sigma^{\text{component}}\begin{pmatrix} s_1 & s_2 \end{pmatrix} + \frac{2}{3}\sigma^{\text{random}}\begin{pmatrix} s_1 & s_2 \end{pmatrix} \\
&= \frac{1}{3}\sigma^{\text{component}}\begin{pmatrix} s_1 & s_2 \end{pmatrix} + \frac{2}{3}\cdot\frac{1}{G}.
\end{aligned} \tag{14}
$$

Thus, to obtain a falling-sticks texture with a given set of correlations, one must choose

$$
\sigma^{\text{component}} = 3\sigma^{\text{falling sticks}} - \frac{2}{G}. \tag{15}
$$

Expressed in terms of distance from randomness, this demonstrates the threefold dilution:

$$
\sigma^{\text{component}} - \frac{1}{G} = 3\left(\sigma^{\text{falling sticks}} - \frac{1}{G}\right). \tag{16}
$$

This limits the maximum correlation strength of the final texture, but it suffices for the present purposes since the achievable correlation strengths are generally far above perceptual threshold. A similar analysis applies to the vertical pairwise correlations.

Note that first-order correlations are zero since the two starting textures had an equal number of checks of all gray levels, and third-order correlations within a $2 \times 2$ block are zero because they always involve checks that originated in independent Markov processes. Some fourth-order correlations are not zero, but their deviations from zero are small: this is because they arise from $2 \times 2$ neighborhoods in which the two last 'sticks' are both horizontal (1/6 of the time) or both vertical (1/6 of the time), and these numerical factors multiply a product of two terms involving $\sigma^{\text{component}} - 1/G$.

Finally, a Metropolis mixing step (*Metropolis et al., 1953*) is applied, to maximize entropy without changing the $2 \times 2$ block probabilities (see *Victor and Conte, 2012* for details). This eliminates any spurious long-range correlations that may have arisen from the 'sticks' of the underlying textures.

## Appendix 3

### Symmetry tests

In order to test to what extent the natural image predictions or the psychophysical measurements are invariant under symmetry transformations, we need to understand the effect of such transformations on texture coordinates $\sigma^{s_1 s_2}_{s_3 s_4}$. Two observations are key. First, every geometric transformation we are interested in—reflections and rotations by multiples of 90°—correspond to permutations of the check locations $A_1$, $A_2$, $A_3$, $A_4$. For instance, a horizontal flip exchanges $A_1$ with $A_2$ and $A_3$ with $A_4$. Second, for the particular case of ternary textures, all the permutations of the three gray levels correspond to affine transformations *modulo* 3. Specifically, given a check value $A$, consider the transformation $A \to xA + y \pmod 3$. $(x, y) = (1, 0)$ is the identity; $(x, y) = (1, 1)$ and $(x, y) = (1, 2)$ are the nontrivial cyclic permutations; and $x = 2$, $y \in \{0, 1, 2\}$ yields the three pairwise exchanges. We exclude $x = 0$ since this would correspond to removing all luminance variations in the image patches.

As we now show, the net result of a geometric transformation and a color permutation always corresponds to a permutation of the texture coordinates. Consider a general permutation $\rho^{-1}$ on the four check locations, $A_k \to A_{\rho^{-1}(k)}$ (it will become clear below why we use the inverse here), and a general affine transformation on the gray levels, $A \to xA + y \pmod G$, with $x \neq 0$. The equation appearing in the definition of the $\sigma^{s_1 s_2}_{s_3 s_4}(h)$ direction (*Equation 10*) becomes:

$$\sum_{k=1}^{4} A_k s_k = h \pmod G \quad \to \quad \sum_{k=1}^{4} (x A_{\rho^{-1}(k)} + y) s_k = h \pmod G. \tag{17}$$

Note that here, $h$ is a label identifying the coordinate direction, and therefore is not transformed by the affine transformation applied to the luminance values $A_k$. Since $x \neq 0$, we find

$$\sum_{k=1}^{4} A_k s_{\rho(k)} = x^{-1} \left( h - y \sum_{k=1}^{4} s_k \right) \pmod G. \tag{18}$$

Thus, the effect of the transformation was to convert the original texture direction into a different one. To properly identify the transformed direction, we need to put it in monic form; that is, we need to ensure that the first non-zero coefficient is set to 1. Suppose that this coefficient appears at position $k_0$, and is equal to $s_{\rho(k_0)}$. We can write:

$$\sum_{k=1}^{4} A_k \tilde{s}_k = s_{\rho(k_0)}^{-1} x^{-1} \left( h - y \sum_{k=1}^{4} s_k \right) \pmod G, \tag{19}$$

where

$$\tilde{s}_k = s_{\rho(k_0)}^{-1} s_{\rho(k)}, \tag{20}$$

and therefore $\tilde{s}_{k_0} = 1$. Thus the direction $\sigma^{s_1 s_2}_{s_3 s_4}(h)$ gets mapped to $\sigma^{\tilde{s}_1 \tilde{s}_2}_{\tilde{s}_3 \tilde{s}_4}(ah - b \pmod G)$ where $a = s_{\rho(k_0)}^{-1} x^{-1}$ and $b = ay \sum s_k$. Thus, all transformations correspond to relabeling of the texture coordinates. This approach holds for any prime $G$, but for $G > 3$ the affine transformations will no longer be sufficient to model all gray level permutations.

To find the effect of the geometric and gray-level transformations on the theoretical predictions, we apply this reshuffling to the columns of the $N_{\text{patches}} \times 99$ matrix giving the distribution of natural image patches in texture space (99 here corresponds to the three probability values in each of the 33 simple texture planes). We then recalculate the threshold predictions and check how much these changed. This is equivalent to first performing the geometric and gray level transformations directly on each image and then rerunning the whole analysis, but is substantially more efficient.

For the psychophysics, we apply the transformation to each direction in texture space for which we have thresholds. In some cases, the transformed direction is not contained in the experimental dataset; we cannot check for symmetry when this happens. We thus only compare the original and transformed thresholds in cases where the transformed direction maps onto one of the directions in the original dataset.

Note that some symmetry transformations leave certain directions in texture space invariant. For instance, a left-right flip leaves the entire $\beta_{11}$ plane invariant since it only flips the order of the terms in the sum $A_1 + A_2$. When this happens, the thresholds obtained in those directions are unaffected by the transformation and thus the fact that they do not change cannot be used as evidence that the symmetry is obeyed. For this reason, all invariant directions are ignored when looking at the effect of a transformation. This explains the variability in the number of points in the different plots from *Figure 6B*.

## Appendix 4

### Statistical tests

We performed several statistical tests to assess the quality of the match that we found between measurements and natural image predictions. For the permutation tests below, we used a scalar measure of the discrepancy that is given by median absolute error calculated in log space, after accounting for an overall scaling factor between psychophysical measurements and natural image predictions. More precisely, suppose we have $n$ measurements $x_i$ and $n$ predictions $y_i$. We define the mean-centered log thresholds

$$\begin{aligned} \hat{x}_i &= \log x_i - \langle \log x_i \rangle, \\ \hat{y}_i &= \log y_i - \langle \log y_i \rangle, \end{aligned} \tag{21}$$

where $\langle . \rangle$ represents the mean over all measurement directions. Then the mismatch between $x$ and $y$ is given by

$$D = \operatorname{median} |\hat{x}_i - \hat{y}_i|. \tag{22}$$

Note that the difference between the natural logarithms of two quantities that are relatively close to each other, $a_1 \approx a_2 \approx \bar{a} \equiv (a_1 + a_2)/2$, is approximately equal to the relative error between the two quantities,

$$\log a_1 - \log a_2 = \log \frac{a_1}{a_2} = \log \left( 1 + \frac{a_1 - a_2}{a_2} \right) = \frac{\Delta a}{a_2} + \mathcal{O}(\Delta a^2), \tag{23}$$

where $\Delta a = a_1 - a_2$, and the big-$\mathcal{O}$ notation shows that the approximation error scales as the square of the absolute prediction error. In fact, the log error is even better approximated by a symmetric form of the relative error, in which the denominator is replaced by the mean between measurement and prediction:

$$\begin{aligned} \log a_1 - \log a_2 &= \log \frac{a_1}{\bar{a}} - \log \frac{a_2}{\bar{a}} = \log \left[ 1 + \frac{a_1 - a_2}{2\bar{a}} \right] - \log \left[ 1 - \frac{a_1 - a_2}{2\bar{a}} \right] \\ &= \frac{\Delta a}{2\bar{a}} - \frac{\Delta a^2}{8\bar{a}^2} + \frac{\Delta a}{2\bar{a}} + \frac{\Delta a^2}{8\bar{a}^2} + \mathcal{O}(\Delta a^3) \\ &= \frac{\Delta a}{\bar{a}} + \mathcal{O}(\Delta a^3). \end{aligned} \tag{24}$$

We see that in this case the approximation error scales with the cubed absolute prediction error. In our case, prediction errors are low enough that the ratio between the log error and the symmetric relative error $\Delta a / \bar{a}$ is between 1 and 1.034 for all 311 thresholds.

1. Permutation test over all thresholds. This test estimates how likely it is that the observed value of the difference $D$ could have been obtained by chance in a model in which all the measured thresholds $x_i$ are drawn independently from a common random distribution. We generated 10,000 random permutations of the dataset $x_i$, $x_i^\mu$ (each of which contain 311 data points in 4 single and 22 mixed planes), and for each of these calculated the difference $D^\mu$ between the observed values and the predictions $y_i$. We then calculated the fraction of samples $\mu$ for which the difference was smaller than the observed one, $D^\mu \leq D$, which is an estimate of the $p$-value.

2. Permutation test preserving ellipticity. The naïve permutation test above generates independent thresholds even for directions that are nearby within a given texture plane. To build a more stringent test where the threshold contours are kept close to elliptical, we used a different sampling procedure where permutations were applied only within texture groups, and were forced to be cyclic. In this way, thresholds obtained for adjacent texture directions were kept adjacent to each other, preserving the correlations implied by the elliptical contours. More specifically, assume that we index the $n_{\text{groups}}$ texture groups for which we have data by $\sigma$, and let $x_\sigma$ be the subset of elements of $x$ corresponding to group $\sigma$. Also let $\mathcal{R}$ be the shift operator that circularly permutes the elements in a vector to the right, such that the first element becomes the second, the second becomes the third and so on, with the last element being moved to the first position. For each resampling of the measurements $x$, we sampled $n_{\text{groups}}$ non-negative integers $k_\sigma$ and performed the transformations $x_\sigma \to \mathcal{R}^{k_\sigma} x_\sigma$. The largest

value for $k_\sigma$ was chosen to be equal to the number of elements in $x_\sigma$. For each of the 10,000 samples obtained in this way we calculated the $D$ statistic and proceeded as above to get a $p$-value.

3. Exponent estimation. The predictions from natural image statistics were obtained by taking the inverse of the standard deviation of the natural texture distribution in each direction. This differs by a square root from the efficient-coding prediction for Gaussian inputs and linear gain (*Hermundstad et al., 2014*). This is not unreasonable since natural scene statistics are not exactly Gaussian, and we do not expect brain processing to be precisely linear. However, to more fully investigate possible non-linearities, we considered general power-law transformations of the predicted thresholds, and used the data to estimate the exponent $\eta$. Specifically, given the mean-centered log predictions $\hat{y}_i$, we asked whether $\eta\hat{y}_i$ is a better approximation to the measurements $\hat{x}_i$ (this implies that $y_i^\eta$ provides a better prediction for the $x_i$ values, which is why we refer to $\eta$ as an exponent). We can write down the model

$$\hat{x}_i = \eta\hat{y}_i + \sigma\epsilon_i, \tag{25}$$

where $\epsilon_i$ are errors drawn from a standard normal distribution, $\epsilon_i \sim \mathcal{N}(0,1)$. This model interpolates between the null model above that assumes all thresholds are drawn from the same distribution (when the exponent $\eta = 0$), and a model in which they are given by the values predicted from natural images, plus noise (with exponent $\eta = 1$). To find $\eta$ and $\sigma$, we used a Bayesian approach. The log posterior distribution for the parameters $\eta$ and $\sigma$ given the measured data $\hat{x}_i$ is:

$$\log P_{\text{posterior}}(\eta, \sigma) = -\frac{1}{2\sigma^2}\sum_i(\eta\hat{y}_i - \hat{x}_i)^2 - n\sigma + \log P(\eta)P(\sigma) + \text{const.}, \tag{26}$$

where $n$ is the number of measurements, and we assumed that $\eta$ and $\sigma$ are independent in our prior distribution. We then used slice sampling (taking advantage of the slicesample function in Matlab, RRID:SCR_001622) to draw from the posterior distribution. We used 10,000 samples in total, and we discarded the first 5000 as a burn-in period. We used a flat prior for $\eta$ and $\log\sigma$, but checked that the results are similar when using other priors.

The results from these statistical tests are summarized in *Appendix 4—table 1* for the various choices of downsampling factor $N$ and patch size $R$, and they show that the match we see holds across the explored range of $N$ and $R$. Interestingly, the 95% credible intervals obtained for the exponent $\eta$ in the exponent-estimation model do not include one for most choices of preprocessing parameters, suggesting that an additional nonlinearity might be at play here.

**Appendix 4—table 1.** Results from statistical tests comparing the match between measured and predicted thresholds to chance.

The left column gives the preprocessing parameters $N$ (the downsampling factor) and $R$ (the patch size) in the format $N \times R$. For each of the permutation tests, a $p$-value and the shortest interval containing 95% of the $D$ values obtained in 10,000 samples is given (the 95% highest-density interval, or HDI). Similarly, for the exponent estimation, we include the shortest interval containing 95% of the posterior density for each of the two model parameters (the 95% highest posterior-density interval, or HPDI).

| | | 1. Permutation #1 | | 2. Permutation #2 | | 3. exponent estimation | |
|---|---|---|---|---|---|---|---|
| (l)3-8 | | **D** | | **D** | | $\eta$ | $\sigma$ |
| $N \times R$ | $D_{\text{actual}}$ | $p$ | [95% HDI] | $p$ | [95% HDI] | [95% HPDI] | [95% HPDI] |
| $1 \times 32$ | 0.20 | $<10^{-4}$ | $[0.26, 0.33]$ | 0.0041 | $[0.21, 0.27]$ | $[0.58, 0.75]$ | $[0.22, 0.26]$ |
| $1 \times 48$ | 0.20 | $<10^{-4}$ | $[0.28, 0.35]$ | 0.0020 | $[0.22, 0.29]$ | $[0.52, 0.67]$ | $[0.22, 0.25]$ |
| $1 \times 64$ | 0.21 | $<10^{-4}$ | $[0.29, 0.37]$ | 0.0010 | $[0.23, 0.31]$ | $[0.50, 0.63]$ | $[0.21, 0.25]$ |
| $2 \times 32$ | 0.13 | $<10^{-4}$ | $[0.24, 0.31]$ | $<10^{-4}$ | $[0.16, 0.22]$ | $[0.81, 0.98]$ | $[0.18, 0.22]$ |

*Continued on next page*

*Appendix 4—table 1 continued*

|  |  | **1. Permutation #1** |  | **2. Permutation #2** |  | **3. exponent estimation** |  |
|---|---|---|---|---|---|---|---|
| (l)3-8 |  | $D$ |  | $D$ |  | $\eta$ | $\sigma$ |
| $2 \times 48$ | 0.15 | $<10^{-4}$ | $[0.26, 0.33]$ | $<10^{-4}$ | $[0.18, 0.23]$ | $[0.71, 0.85]$ | $[0.18, 0.21]$ |
| $2 \times 64$ | 0.16 | $<10^{-4}$ | $[0.27, 0.34]$ | $<10^{-4}$ | $[0.18, 0.24]$ | $[0.66, 0.79]$ | $[0.18, 0.22]$ |
| $4 \times 32$ | 0.12 | $<10^{-4}$ | $[0.24, 0.30]$ | $<10^{-4}$ | $[0.15, 0.20]$ | $[0.87, 1.04]$ | $[0.18, 0.22]$ |
| $4 \times 48$ | 0.14 | $<10^{-4}$ | $[0.26, 0.33]$ | 0.0003 | $[0.16, 0.22]$ | $[0.74, 0.89]$ | $[0.18, 0.21]$ |
| $4 \times 64$ | 0.15 | $<10^{-4}$ | $[0.26, 0.34]$ | 0.0002 | $[0.16, 0.23]$ | $[0.69, 0.83]$ | $[0.18, 0.21]$ |

## Appendix 5

### Systematic trends in the prediction errors

Overall, the natural-image predictions are in good agreement with the measured psychophysical thresholds. The discrepancies, when they occur, are not entirely random. These trends are shown in *Figure 1*. As in the main text, we focus on second-order correlations.

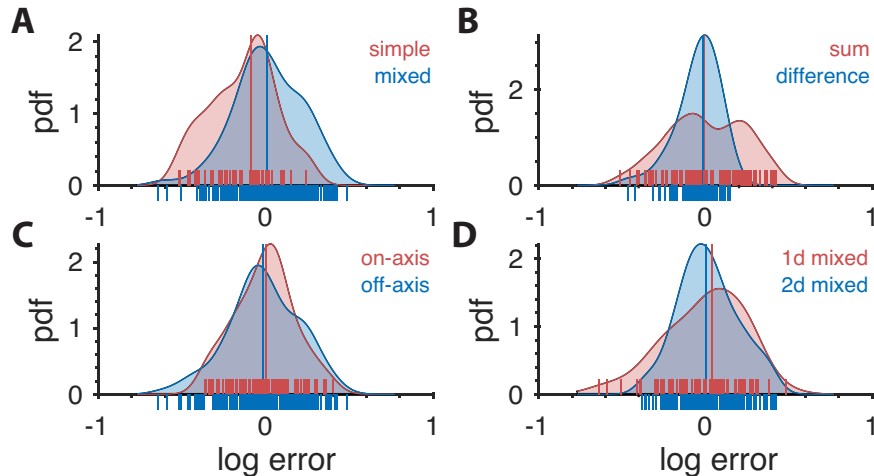

**Appendix 5—figure 1.** Distribution of prediction errors across specific subsets of thresholds. Each plot shows kernel-density estimates of the distribution of log prediction errors (defined as difference between log prediction and log measurement) after splitting the data into the subgroups indicated at the top right of each plot. Individual data points are shown on the abscissa. The median log prediction error is shown for each group in the corresponding color. (**A**) Predictions tend to underestimate thresholds in simple planes and overestimate thresholds in mixed planes ($p = 0.0014$, Kolmogorov-Smirnov (K–S) test). As a reminder, the natural-image analysis predicts thresholds only up to a multiplicative factor, which is chosen in a way that makes the mean log prediction error over all second-order thresholds be zero. (**B**) Predictions tend to have greater error in planes defined by modular sums (such as the simple plane $\beta_{++}$ or the mixed plane $(\beta_{++}[0], \beta_{\substack{+ \\ +}}[1])$) than in planes defined by modular differences (such as the simple plane $\beta_{+-}$ or the mixed plane $(\beta_{+-}[1], \beta_{\substack{+ \\ -}}[0])$) ($p = 1.8 \cdot 10^{-4}$, K-S test). However, thresholds in neither subgroup are systematically over- or underestimated. (**C**) There is no significant difference in predictions in on-axis directions (directions parallel to an axis in a simple or mixed coordinate plane), *vs.* all other (off-axis) directions ($p = 0.56$, K-S test). (**D**) Thresholds are more accurately predicted for '2-D' correlations than '1-D' correlations. A mixed plane like $(\beta_{++}[0], \beta_{+-}[0])$ involves the same pair of checks in both directions, thus leading to correlations that are in a sense 1-D. In contrast, the mixed plane $(\beta_{++}[1], \beta_{\substack{+ \\ -}}[0])$ involves the three checks $A_1$, $A_2$, and $A_3$, leading to 2-D correlations. While the medians of the errors in these two subgroups are similar (KS test $p = 0.25$; Wilcoxon rank-sum test $p = 0.96$, medians 0.042 *vs.* 0.004, respectively), prediction error magnitude is lower for 2-D correlations ($p = 0.01$, KS-test on absolute log errors).

### Simple-plane thresholds are more often underestimated than mixed-plane thresholds

As shown in *Figure 1A*, prediction errors in simple planes tend to be negative, while prediction errors in mixed planes tend to be slightly positive (medians $-0.090$ and $+0.008$; $p = 0.0014$, Kolmogorov-Smirnov (K-S) test). Note that on balance, prediction errors will be close to zero, since we applied an overall scale factor to minimize the overall difference between predictions from natural images and psychophysical data. Thus, this finding means that perceptual performance when more than one kind of correlation is present is (slightly) disproportionately better than perceptual performance when only one kind of correlation is present. This suggests that the brain uses

more resources to analyze mixed-plane correlations than our efficient-coding model would predict, or that processing of multiple correlations is in some sense synergistic.

## Sum directions tend to have higher errors than difference directions

Second-order correlations in the texture space employed here are defined using either sums or differences (modulo 3) of discretized luminance values. The wrap-around due to the modular arithmetic has different effects on these two kinds of correlations. Specifically, correlations due to modular sums imply a tendency for adjacent checks of a specific gray level to match (see main text *Figure 1C*), while correlations due to modular differences imply a tendency for checks to match their neighbor independent of gray level, *or*, for mini-gradients (black to gray to white) to be present (see main text *Figure 1D*). To test whether this distinction influenced prediction error, we split the data into planes (either simple or mixed) that use only modular sums, and planes (simple or mixed) that use only modular differences. Mixed planes such as $(\beta_{++}[0], \beta_{+-}[0])$ that use both sums and differences were excluded from this analysis.

*Figure 1B* shows that while there is no significant difference between the median log errors in sum *versus* difference directions, the full distributions are different (K-S $p$-value is $p = 1.8 \cdot 10^{-4}$), largely due to the fact that the sum directions exhibit significantly larger absolute errors compared to the difference ones (median absolute log error 0.19 *vs.* 0.07, respectively). We suggest that this reflects the greater dependence of the sum-correlations on the gray-level discretization.

## On-axis and off-axis prediction errors are roughly matched

We next examined whether predictions were more or less accurate on the coordinate axes, by comparing prediction errors for on-axis correlations (i.e., correlations described by one of the $p_i = 1$ corners of the probability simplex (see *Figure 1*); or one of the two axes in mixed planes [see *Figure 2*]) with all other measurements (off-axis). The distributions of prediction errors for on-axis *vs.* off-axis thresholds do not exhibit any noticeable differences (see *Figure 1C*).

## Different kinds of mixed planes have slightly different error magnitudes

Finally, we noted that in the mixed planes, some correlations involved only two checks—and thus were one-dimensional, while others involved overlapping dyads in orthogonal directions and thus were two-dimensional. Specifically, in a mixed plane such as $(\beta_{++}[0], \beta_{+-}[0])$, only a horizontally-oriented pair of checks is involved in both types of correlations, meaning that nearby rows in such textures are independent. In contrast, in the mixed plane $(\beta_{++}[1], \beta_{+}[0])$, a horizontal pair of checks is used in the first dimension and a vertical one in the second, leading to correlations along both dimensions. This distinction was not associated with over- or under-prediction of thresholds (KS test $p = 0.25$; Wilcoxon rank-sum test $p = 0.96$, medians 0.042 and 0.004, respectively), but the distribution of errors was wider for the 1-D correlations than for the 2-D correlations (0.19 *vs.* 0.12, $p = 0.01$, K-S test on absolute log errors).

**Large thresholds are underestimated**

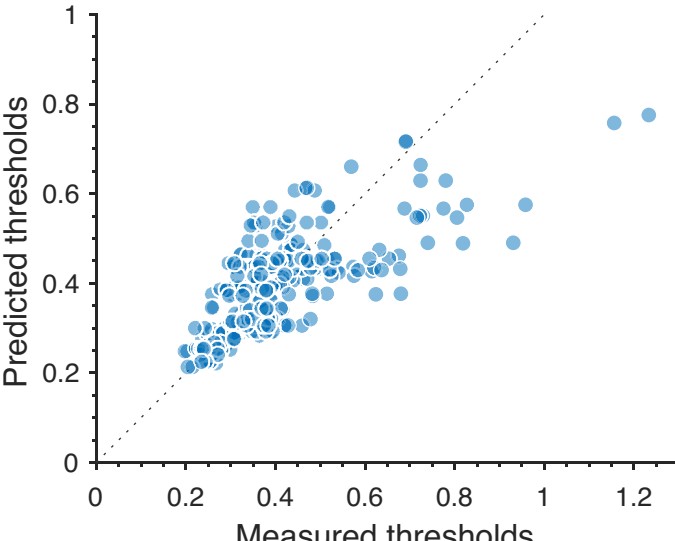

**Appendix 5—figure 2.** Predictions for large thresholds, corresponding to low-sensitivity directions in texture space, consistently underestimate measured thresholds. The plot shows the 311 second-order values.

In directions where human subjects showed low sensitivity to textures, leading to large measured thresholds, the natural-image predictions were consistently lower than the measurements. This suggests that adding a nonlinearity to the model could improve the fit to the data. Indeed, as shown in Appendix 4, when fitting a model of the form $\mathrm{threshold} \propto (\mathrm{standard\ deviation})^{-\eta}$, an exponent slightly smaller than one is most consistent with the data (95% credible interval for $\eta$ is $[0.81, 0.98]$).

## Appendix 6

### Robustness tests

The match between psychophysical thresholds and efficient coding predictions is robust under a number of alterations in the analysis pipeline, including changing the parameters of the preprocessing and using different image databases.

### Changing preprocessing parameters

As we showed in the main text, changing the patch size $R$ or the downsampling ratio $N$ does not significantly affect the match between experiment and theory. Another aspect of the preprocessing that we have varied is the ternarization procedure. In the main text, images were ternarized by splitting the entire dynamic range of each patch into three regions, each corresponding to an equal number of checks (up to one check, due to the fact that our patch sizes are not divisible by three). Here, we consider variants of the ternarization procedure, parameterized by the fraction $\rho$ of checks that are mapped to gray, while assuming that the remaining $1 - \rho$ checks are equally distributed between black and white. Thus, the default ternarization procedure corresponds to $\rho = 1/3$. The effects of varying $\rho$ are small in most texture planes, with an increase in prediction error as we move away from histogram equalization ($\rho = 1/3$), as seen in the figure below.

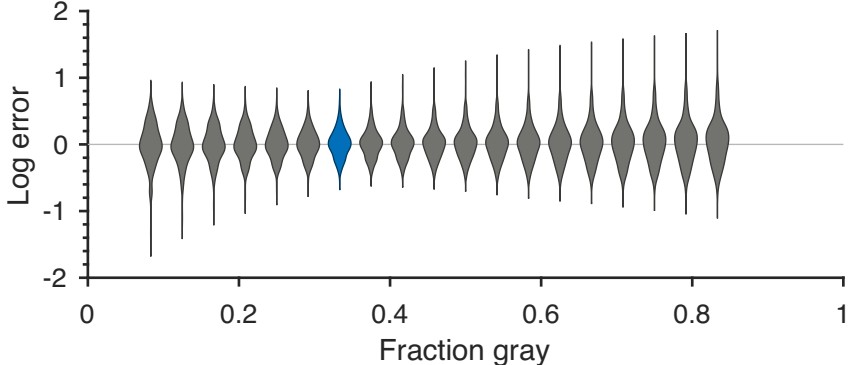

**Appendix 6—figure 1.** Robustness to changing ternarization thresholds. We varied the fraction of gray checks in the ternarized natural image patches, while keeping the fractions of black and white checks equal to each other. For each value of the fraction of gray checks, we recalculated the threshold predictions and compared them to the psychophysical measurements. Each violin in the figure shows a kernel density estimate for the distribution of prediction errors (in log space) for the 311 second-order single- and mixed-plane threshold measurements available in the psychophysics. We see that the precise thresholds used for ternarization do not significantly affect the match between natural image predictions and psychophysics. The lowest error is close to the value 1/3 which corresponds to equal fractions of black, gray, and white checks, and is the one used in the main text. The corresponding violin is highlighted in blue in the figure.

### Changing the image database

The accuracy of the natural image threshold predictions is also good when we use different image databases. *Figure 2* shows the match between predictions and measurements in each plane when using the van Hateren database (*van Hateren and van der Schaaf, 1998*) with a downsampling ratio $N = 2$ and patch size $R = 32$.

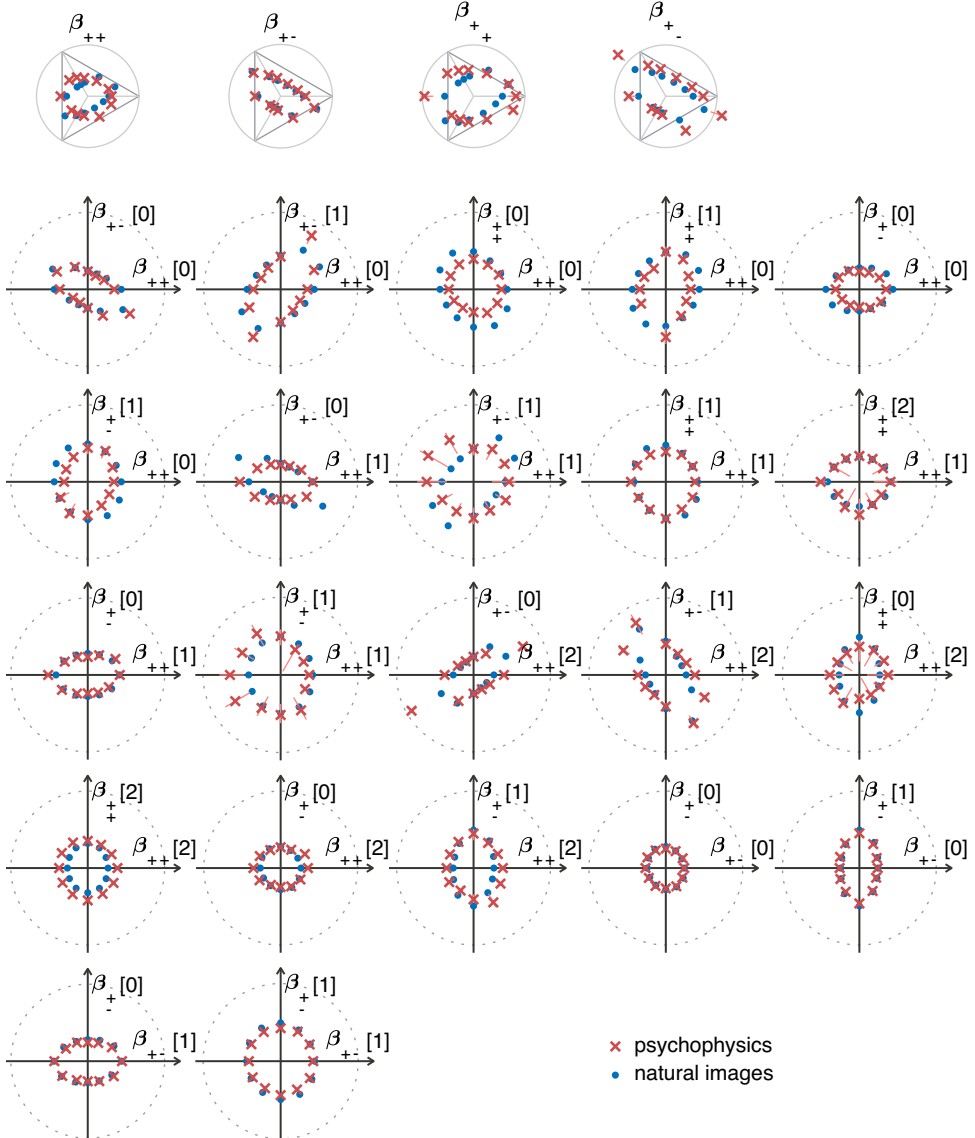

**Appendix 6—figure 2.** Natural image predictions (blue dots) for second-order planes when using the van Hateren image database (*van Hateren and van der Schaaf, 1998*). The psychophysics measurements are also shown, in red crosses. The notations are as described in the main text (*Figure 5*).

The distance measure from *Equation (22)* comparing these results to the results obtained from the Penn Image Database using the same preprocessing options was $D_{\text{Penn}-\text{vH}} = 0.086$, showing that the two sets of measurements not only both agree with the psychophysics, but also agree with each other. *Appendix 6—table 1* shows the results of the statistical tests for various preprocessing parameters for the van Hateren database.

**Appendix 6—table 1.** Results from statistical tests comparing the match between measured and predicted thresholds to chance when using the van Hateren natural image database (*van Hateren and van der Schaaf, 1998*).

The left column gives the preprocessing parameters $N$ (the downsampling factor) and $R$ (the patch size) in the format $N \times R$. For each of the permutation tests, a $p$-value and the shortest interval containing 95% of the $D$ values obtained in 10,000 samples is given (the 95% highest-density interval, or HDI). Similarly, for the exponent estimation, we include the shortest interval containing 95% of the posterior density for each of the two model parameters (the 95% highest posterior-density interval, or HPDI).

| (l)3-8 | | **1. Permutation #1** | | **2. Permutation #2** | | **3. exponent estimation** | |
|---|---|---|---|---|---|---|---|
| | | *D* | | *D* | | η | σ |
| $N \times R$ | $D_{\text{actual}}$ | $p$ | [95% HDI] | $p$ | [95% HDI] | [95% HPDI] | [95% HPDI] |
| $1 \times 32$ | 0.22 | $<10^{-4}$ | [0.27, 0.34] | 0.0081 | [0.23, 0.31] | [0.45, 0.63] | [0.24, 0.28] |
| $1 \times 48$ | 0.23 | $<10^{-4}$ | [0.30, 0.37] | 0.0013 | [0.26, 0.34] | [0.41, 0.57] | [0.24, 0.28] |
| $1 \times 64$ | 0.24 | $<10^{-4}$ | [0.30, 0.38] | 0.0020 | [0.26, 0.34] | [0.40, 0.54] | [0.24, 0.27] |
| $2 \times 32$ | 0.16 | $<10^{-4}$ | [0.26, 0.32] | 0.0001 | [0.19, 0.25] | [0.67, 0.84] | [0.21, 0.24] |
| $2 \times 48$ | 0.18 | $<10^{-4}$ | [0.27, 0.34] | 0.0003 | [0.20, 0.27] | [0.58, 0.73] | [0.21, 0.24] |
| $2 \times 64$ | 0.19 | $<10^{-4}$ | [0.28, 0.36] | 0.0002 | [0.21, 0.28] | [0.54, 0.67] | [0.21, 0.24] |
| $4 \times 32$ | 0.14 | $<10^{-4}$ | [0.25, 0.32] | 0.0002 | [0.17, 0.24] | [0.73, 0.90] | [0.20, 0.23] |
| $4 \times 48$ | 0.15 | $<10^{-4}$ | [0.27, 0.34] | $<10^{-4}$ | [0.18, 0.26] | [0.64, 0.78] | [0.19, 0.23] |
| $4 \times 64$ | 0.16 | $<10^{-4}$ | [0.28, 0.36] | $<10^{-4}$ | [0.20, 0.26] | [0.59, 0.73] | [0.19, 0.23] |

## Subject dependence

As mentioned in the main text, the psychophysical thresholds show remarkable consistency across subjects, with most differences attributable to an overall scaling factor. The results are shown in *Figure 3*.

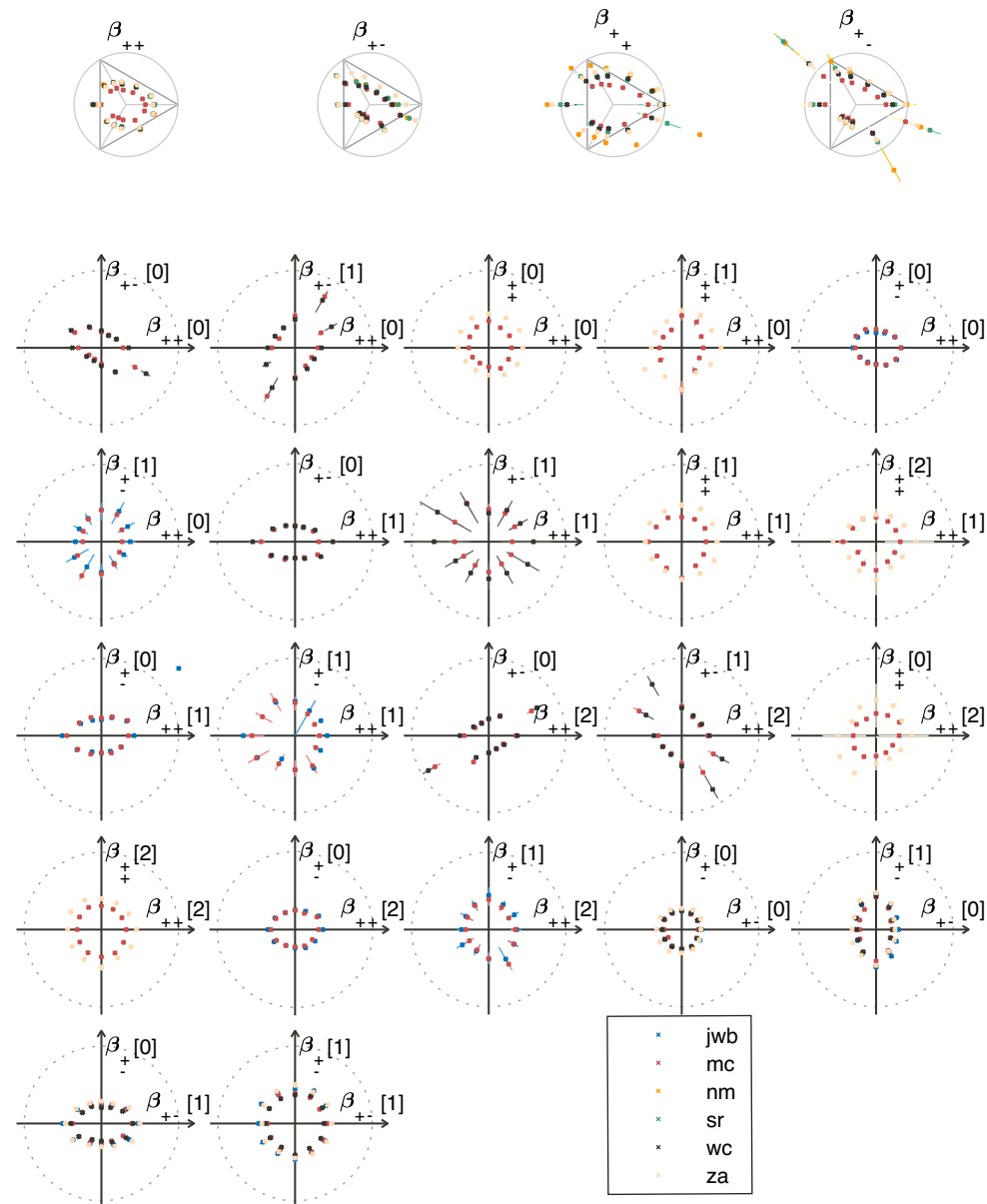

**Appendix 6—figure 3.** Psychophysical thresholds in the second-order planes shown for different subjects. Depending on the plane, measurements were made in 2–5 subjects in each texture direction. The notations are as in the main text (*Figure 5*).

The differences between different subjects were more pronounced in higher-order planes (*Figure 4*). Note that the predicted thresholds (shown as blue dots in the figure) are still not far from the measurements; in particular, the predictions reflect the fact that the thresholds in the higher-order planes are generally high.

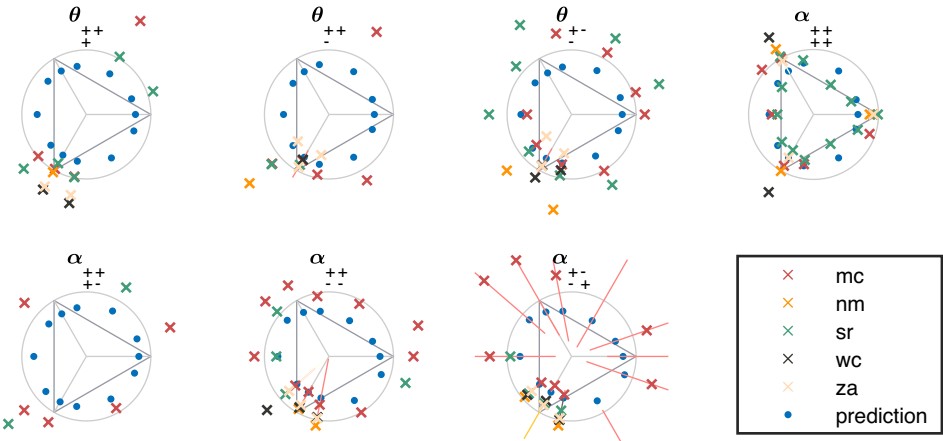

**Appendix 6—figure 4.** Psychophysical thresholds in higher-order planes for individual subjects (crosses). The natural image predictions are also shown, in blue dots. The notations are as in the main text (*Figure 4*). in many directions, performance did not sufficiently exceed chance to allow for a reliable determination of threshold; in these cases, data points are omitted.

