## [Decision Letter]

Thank you for submitting your article "Sensitivity to grayscale textures is adapted to natural scene statistics" for consideration by *eLife*. Your article has been reviewed by two peer reviewers, and the evaluation has been overseen by a Reviewing Editor and Timothy Behrens as the Senior Editor. The reviewers have opted to remain anonymous.

The reviewers have discussed the reviews with one another and the Reviewing Editor has drafted this decision to help you prepare a revised submission.

Summary:

In this manuscript, Tesileanu and co-authors describe a new set of psychophysical experiments and analyses that ask how perceptual salience is related to natural scene variability, specifically looking at the statistics of pairs of ternary pixel values. They find strong concordance between perceptual threshold and the variability of statistics in natural scenes, so that lower detection thresholds exist for the more variable directions in their defined statistical space. This work is solid and follows nicely on their prior work published in *eLife*, making it a good Research Advance.

Essential revisions:

1) Can something be said about which directions showed the biggest differences between psychophysics and natural scenes? In the Discussion, it's stated that it's hard to be perfect and low variance dimensions are difficult. But is there any more specific significance to which patch patterns deviate most from this variance hypothesis?

2) Please expand on how to interpret these results in light of the whitening process, which removed the average pairwise correlations from the image patches. The idea is that deviations from the mean 2-point correlations result in saliency, but is it troubling that the psychophysical stimulus itself did not have the mean 2-point correlations that exist in natural scenes? How would the visual system be exquisitely sensitive to deviations in natural scenes, but still work identically even in the absence of naturalistic pairwise correlations in the psychophysical stimulus? It seems as though this absence of naturalistic pairwise correlations would mean that the observer's visual system is pretty far from where it might reside under realistic natural viewing conditions. Yet the variability still predicts salience. Is this notable or is this easily explained?

3) Enhance the clarity of the Results section:

The theoretical framework set up here is very nice, but challenging to grasp for a naive reader. Many details are left to the appendix that could be moved to the main text (as a summary) to improve the clarity of the presentation. On the other hand, there are some details in the Results that detract from the clarity of the presentation that could be moved to the Materials and methods. Please consider these points and revise the Results section accordingly.

---

## [Author Response]

Essential revisions:1) Can something be said about which directions showed the biggest differences between psychophysics and natural scenes? In the Discussion, it's stated that it's hard to be perfect and low variance dimensions are difficult. But is there any more specific significance to which patch patterns deviate most from this variance hypothesis?

We thank the reviewers for raising this point, and pursuing it reveals two interesting trends. This analysis is detailed in a new Appendix and summarized in the text (see Appendix 5, and subsection “Variance predicts salience”).

First, predictions in simple planes tend to underestimate actual thresholds, while predictions in mixed planes tend to overestimate actual thresholds (effect size of about 9%). As we elaborate in the appendix, this suggests that the brain puts more resources into analyzing mixed-plane correlations than our efficient-coding model would predict.

A second observation is that prediction errors are, on average, almost three times larger in directions defined by modular sums (like 𝛽_++_) than those defined by modular differences (like 𝛽_+-_), namely, 19% vs. 7%. As explained in the new material, we suggest that this reflects a kind of non-robustness (e.g., sensitivity to discretization) associated with the modular-sum correlations that distinguishes them from the modular-difference correlations.

2) Please expand on how to interpret these results in light of the whitening process, which removed the average pairwise correlations from the image patches. The idea is that deviations from the mean 2-point correlations result in saliency, but is it troubling that the psychophysical stimulus itself did not have the mean 2-point correlations that exist in natural scenes? How would the visual system be exquisitely sensitive to deviations in natural scenes, but still work identically even in the absence of naturalistic pairwise correlations in the psychophysical stimulus? It seems as though this absence of naturalistic pairwise correlations would mean that the observer's visual system is pretty far from where it might reside under realistic natural viewing conditions. Yet the variability still predicts salience. Is this notable or is this easily explained?

During natural vision, a substantial contribution to whitening of the visual input comes from fixational eye movements (Rucci and Victor, 2015). Spatial filtering has also been thought to play a role (Atick and Redlich, 1990), but in vitro experiments (Simmons et al., 2013) found that adaptive spatiotemporal receptive field processing did not by itself whiten the retinal output, but rather served to maintain a similar degree of correlation across stimulus conditions. We infer from these studies that short visual stimuli like our 120ms presentations should be pre-whitened, to make up for the absence of fixational eye movements that produce whitening in natural, continuous viewing conditions. We have now commented on this in the subsection entitled “Psychophysical Measurements”.

3) Enhance the clarity of the Results section:The theoretical framework set up here is very nice, but challenging to grasp for a naive reader. Many details are left to the appendix that could be moved to the main text (as a summary) to improve the clarity of the presentation. On the other hand, there are some details in the Results that detract from the clarity of the presentation that could be moved to the Materials and methods. Please consider these points and revise the Results section accordingly.

We made several changes to enhance clarity and streamline the main text:

– We outlined the procedure used for generating texture patches in the first subsection of the Results (subsection “Local textures with multiple gray levels”)

– We expanded the legend for Figure 2

– We moved the preprocessing details from the Results section to the Materials and methods (subsection “Local textures with multiple gray levels”)

– We updated the legend for Figure 5 to match the similar explanation from Figure 2’s legend

– We added an example of how we can infer the effects of a symmetry transformation on our threshold prediction (subsection “Invariances in psychophysics recapitulate symmetries in natural images”)

We also simplified the statistical analysis (not specifically requested, but we think this helps):

– We replaced the relative error / relative change measures that we had used in Figure 6 by log errors, for consistency with the rest of the paper

– We replaced the root-mean-square measure *D* that we were using for the statistical tests with a median, again for consistency and for more robustness to outliers (Appendix 4)

– We added summary statistics and 95% confidence intervals in addition to *p*-values in the Results section (subsection “Variance predicts salience”)